# Size distribution and ionic composition of marine summer aerosol at the continental Antarctic site Kohnen

Rolf Weller[1], Michel Legrand[2], and Susanne Preunkert[2]

[1]Alfred Wegener Institute for Polar and Marine Research, Am Handelshafen 12, D-27570 Bremerhaven, Germany

[2]Université Grenoble Alpes, CNRS, Laboratoire de Glaciologie et Géophysique de l'Environnement (LGGE), Grenoble, France

*Correspondence to*: R. Weller (rolf.weller@awi.de)

**Abstract.** We measured aerosol size distributions and conducted bulk as well as size segregated aerosol sampling during two summer campaigns in January 2015 and January 2016 at the continental Antarctic station Kohnen (Dronning Maud Land). Physical and chemical aerosol properties differ conspicuously during the episodic impact of an distinctive low pressure system in 2015 (LPS15) compared to the prevailing clear sky conditions: The about three days persisting LPS15, located in the eastern Weddell Sea, was associated with marine boundary layer air mass intrusion, enhanced condensation particle concentrations ($1400\pm700$ cm$^{-3}$ compared to $250\pm120$ cm$^{-3}$ under clear sky conditions; mean $\pm$ SD), occurrence of a new particle formation event exhibiting a continuous growth of particle diameters ($D_p$) from 12 nm to 43 nm over 44 hours (growth rate 0.6 nm h$^{-1}$), peaking methane sulfonate (MS$^-$), non–sea salt sulfate (nss–SO$_4^{2-}$) and Na$^+$ concentrations (190 ng m$^{-3}$ MS$^-$, 137 ng m$^{-3}$ nss–SO$_4^{2-}$, and 53 ng m$^{-3}$ Na$^+$ compared to $24\pm15$ ng m$^{-3}$, $107\pm20$ ng m$^{-3}$ and $4.1\pm2.2$ ng m$^{-3}$, respectively, during clear sky conditions, and finally an increased MS$^-$/nss–SO$_4^{2-}$ mass ratio ß$_{MS}$ of 0.4 up to 2.3 ($0.21\pm0.1$ under clear sky conditions) comparable to typical values found at coastal Antarctic sites. Throughout the observation period a larger part of MS$^-$ could be found in super micron aerosol compared to nss–SO$_4^{2-}$, i.e. $(10\pm2)$ % by mass compared to $(3.2\pm2)$ %, respectively. On the whole, under clear sky conditions aged aerosol characterized by usually mono–modal size distribution around $D_p = 60$ nm was observed. Although our observations indicate that sporadic impacts of coastal cyclones were associated with enhanced marine aerosol entry, at large aerosol deposition on–site during austral summer should be dominated by the typical steady clear sky conditions.

## 1 Introduction

The impact of aerosols on global climate, which is in particular mediated by governing cloud droplet concentrations and hence cloud optical properties (Rosenfeld et al., 2014; Seinfeld et al., 2016), is of crucial importance but likewise notoriously charged with the largest uncertainties (Boucher et al., 2013; Seinfeld et al., 2016). In a seminal review Carslaw et al. (2013) concluded that uncertainties in cloud radiative forcing are inter alia caused by uncertainties in natural emissions of dimethylsulfide (DMS) producing biogenic sulfur aerosol (i.e. $MS^-$ and $nss–SO_4^{2-}$) and sea spray. Especially the radiation budget over the Southern Ocean is a challenge for current climate models, as they disturbingly underpredict the aerosol optical depth, pointing to a missing source of aerosols influencing cloud properties in this region (Bodas-Salcedo et al., 2014; Humphries et al., 2016). The competing role of biogenic sulfur (linked with phytoplankton presence) and otherwise sea salt aerosol (linked with stormy sea) in controlling cloudiness above the Southern Ocean is still up for debate (Meskhidze and Nenes, 2006; Korhonen et al., 2008; Quinn and Bates, 2011; Gras and Keywood, 2017).

Biogenic sulfur aerosol, i.e. secondary aerosol produced by photo oxidation of DMS as well as primary sea salt aerosol dominate by far the aerosol mass over the Southern Ocean around Antarctica (Raes et al., 2000; Quinn and Bates, 2011). This original marine aerosol is partly transported to continental Antarctica and eventually deposited on the ice shield. Ideally, deposited aerosol constituents are archived in chronological order in firn (densified snow) and ice (Legrand and Mayewski, 1997). Therefore ice core records of biogenic sulfur and sea salt tracers potentially provide invaluable information on their (strictly speaking local) atmospheric budget which is intrinsically tied to the Southern Ocean climate in the past (Wolff et al., 2006; Kaufmann et al., 2009; Mayewski et al., 2009; Abram et al., 2013). More specifically, investigations on sea salt and biogenic sulfur records from the EPICA (European Project for Ice Coring in Antarctica) ice core retrieved at Kohnen station in Dronning Maud Land revealed the relationship of these archived aerosol tracer profiles with climate indices as there are the Antarctic Circumpolar Wave or the Antarctic Dipol (Fischer et al., 2004; Fundel et al., 2006). In view of the poor knowledge on aerosol source strength and atmospheric concentrations regarding the Southern Ocean realm, retrieving respecting representative historic data from ice core archives is consequently desirable. Certainly, any meaningful interpretation of ice core records rely on the knowledge of the source region and major transport processes as well as transport efficiency to continental Antarctica which is connected with the instant general weather situation. These crucial points can only be elucidated by dedicated aerosol measurements on–site.

In this way, previous aerosol investigations (bulk and sometimes size–segregated composition) revealed a striking difference in the seasonality of sulfur aerosol composition between coastal and inland Antarctica, with $MS^-/nss–SO_4^{2-}$ mass ratios ($\beta_{MS}$) reaching a maximum in January at the coast (0.40 at Neumayer for instance, Legrand and Pasteur (1998)) contrasting with mid–summer $\beta_{MS}$ as low as 0.2 or less at inland Antarctic sites including the South Pole (Arimoto et al., 2004) and the two deep ice core drilling sites of the EPICA project (Kohnen, Weller and Wagenbach (2007) and Concordia, Preunkert et al. (2008) and Becagli et al. (2012)). Based on an extended (2006–2015) records of aerosol, Legrand et al. (2017a) found that low values of $\beta_{MS}$ in mid–summer at Concordia ($0.05 \pm 0.02$) coincided with periods of high photochemical activity as indicated by the presence of locally photo–chemically produced $O_3$. This outcome strongly suggests the occurrence of an efficient chemical destruction of $MS^-$ over the Antarctic plateau in mid–summer. In addition dedicated aerosol investigations within EPICA at Kohnen (Piel et al., 2006) have demonstrated amongst others the conspicuous impact of a cyclone on aerosol transport: In the aftermath of an intense low pressure system over the eastern Weddell Sea region in combination with a blocking high pressure ridge to the East in early January 2002 (Birnbaum et al., 2006), $MS^-$ and $nss–SO_4^{2-}$ aerosol entry showed an exceptional maximum (Piel et al., 2006). So far, a distinct impact of cyclones on aerosol concentrations has been frequently observed at coastal sites (e.g. Ito and Iwai, 1981 for Syowa and Pant et al., 2010 for Maitri), but only rarely on the Antarctic Plateau region (Hogan and Barnard, 1978 for South Pole). In the most elaborate study about this topic, Pant et al. (2010)

reported on almost constant background aerosol concentration under calm conditions, while during the passage of cyclones coarse mode sea salt aerosol increased by an order of magnitude and occasionally new particle formation could be detected in the aftermath of the storms.

Up to now, all investigations at Kohnen were purely based on bulk aerosol sampling. The aim of our present study now focussed on the variability of aerosol number concentration, aerosol size distribution, and size segregated aerosol sampling to investigate the impact of different general weather situations on the physical and chemical properties of the aerosol for a site located on the Antarctic Plateau. This extended synoptic documentation of physics and chemistry of Antarctic aerosol primarily concentrates here on biogenic sulfur aerosol, due to its distinct seasonal summer concentration peak caused by the seasonality of marine biogenic activity of the surrounding Southern Ocean (Weller and Wagenbach, 2007). Hence deposition in Dronning Maud Land should be virtually entirely governed by the atmospheric concentration maximum on–site, which was sufficiently covered by our dedicated observations.

## 2 Experimental techniques and data evaluation methods

### 2.1 Site description

During two summer seasons between 16 Jan. and 3 Feb. 2015 and 12 Jan. and 29 Jan. 2016, we conducted aerosol size distribution measurements and size segregated aerosol samplings at the continental Kohnen station (75°00'S, 00°04'E, altitude: 2892 m a.g.l.; http://www.awi.de/en/expedition/stations/kohnen-station.html, last access: 7 December2017). This summer camp is the former deep ice core drilling location within the EPICA project in Dronning Maud Land about 550 km away from the ice shelf edge. Apart from the given internet link, another detailed description of the site, comprising technical issues as well as a retrospect of the scientific activities can be found in Oerter et al. (2009). In order to minimize the impact of contamination from the permanently running diesel generator at the station, all experiments were installed inside (in situ measurements) or around (aerosol sampling) a bivouac hut located in the clean air sector about 250 m to the northeast of Kohnen (Fig. 1). The power supply (7 kW) was provided by cable from the main station. No fuel driven generator was operated in the designated clean air sector (Fig. 1) and motor vehicle traffic was strictly prohibited. Contamination–free sampling was controlled by the permanently recorded wind velocity and direction. Contamination was indicated for each of the following criteria: Wind direction within a 260°–340° sector and/or wind velocity <1.5 m s$^{-1}$. In case of contamination, given by these criteria, we interrupted the aerosol sampling experiments (low volume and impactor).

For convenience we will use throughout this work the notion day of the year (doy) instead of the calendar date and time is indicated in UTC, which is virtually identical to the solar time. All trace compound concentrations refer to standard volume at 273°K and 1013 hPa (STP).

### 2.2 Experimental set–up

#### 2.2.1 Particle concentration and size distribution

An overview of the experimental set–up during both campaigns is given in Table 1, comprising the respecting measuring periods of the different configurations as well as the relevant specifications of the deployed instruments. The size distribution of the sub–µm aerosol at Kohnen was determined by a scanning mobility particle sizer (SMPS, TSI classifier model 3080; Wang and Flagan (1990)). Set–up and respecting data evaluation methods were virtually identical to our installation run at Neumayer Station, already described in detail elsewhere (Weller et al., 2015). Hence we simply highlight here the most relevant points. The SMPS was generally run in combination with one and the same condensation particle counter (TSI model WCPC 3788, 50% cut–off diameter $D_{p(50\%)}$ of 2.5 nm) controlled by the TSI software AIM (Aerosol Instrument Manager®, version

9.0). The classifier was operated alternately with a so–called nano–DMA (nano differential mobility analyser, TSI Model 3085) and a DMA model 3081. Equipped with the nano–DMA 3085, we adjusted aerosol and sheath flow of the classifier to achieve nominal aerosol size distribution measurements between 2 nm and 64 nm, while in use with the DMA 3081, flows were adjusted to cover the size range between 10 nm and 420 nm, in either cases with 64 channel resolution. Both DMA types were operated with a scanning time of 120 s (retrace time 15 s) and the average size distribution of 4 consecutive (multiple charge and diffusion corrected) spectra was finally evaluated, resulting in a temporal resolution of 600 s.

Particle size distributions were complemented with continuous condensation particle (CP) concentration measurements (TSI, model CPC 3775, $D_{p(50\%)}$ of 4 nm in 2015 and CPC 3025A, $D_{p(50\%)}$ of 3 nm in 2016). Below, we will use the terms nucleation mode for particles with $D_p$ <25 nm and Aitken mode for the size range between 25 nm and 100 nm (Dal Maso et al., 2005).

**2.2.2 Low volume and impactor aerosol sampling**

The low volume sampler (ISAP® 1050, Schulze Automatisierungstechnik, Germany) and the 5–stage low pressure Berner type impactor (GIV, model 80/0.05/2.88) were installed outdoors ca. 15 m northeast of the bivouac hut with ambient air inlets about 1.70 m above ground. Low volume sampling (1.66 m³ h⁻¹ (STP), face velocity 0.33 m s⁻¹) was regularly conducted in 24–hour time periods, while the impactor was run with a flow rate of 4.74 m³ h⁻¹ (STP) in 2–day or 3–day intervals (Table 1; sample exchange usually in the early afternoon around 14:00). For bulk aerosol sampling with the low volume sampler we used a Teflon and a nylon (Nylasorb) filter in series (Pall Corp., all 47 mm diameter and 1 µm pore size). The low volume sampling procedure and filter handling was the same as in our Kohnen campaigns in 2001 and 2002, described in Piel et al. (2006). The impactor was equipped with tedlar foils (Hauke GmbH, Austria) and the aerodynamic cut–off of the stages were corrected for ambient conditions according to Stokes law (Table 1).

According to Piel et al. (2006) sample extracts were analyzed by ion chromatography (IC) for methane sulfonate ($CH_3SO_3^-$, MS⁻), Cl⁻, Br⁻, $NO_3^-$, $SO_4^{2-}$, Na⁺, $NH_4^+$, K⁺, $Mg^{2+}$, and $Ca^{2+}$. The nss–$SO_4^{2-}$ part was calculated using the sea salt tracer Na⁺ according to Piel et al. (2006). Unfortunately, due to unbearable nitrate blanks (caused by an unperceived and unauthorized use of nitric acid in the IC laboratory), all measured $NO_3^-$ concentrations were treated as meaningless and discarded. In addition, especially Br⁻ but also K⁺, $Mg^{2+}$, and $Ca^{2+}$ concentration were frequently close to the analytical detection limit (DL) of around 0.2 ng m⁻³. Figure 2 summarizes the overall experimental uncertainty and ascertained DL, valid for the main ionic compounds to be discussed below. The circles are representative percentage error at different concentrations above the given DL ($\xi_i$), considering both, the analytical (filter handling and IC) accuracy and the uncertainty of the sampling volume (<±5%). By this procedure the error close to the DL was typically ±25% for all ions. The DL was derived from dedicated procedure blanks (in total 10 blanks for low volume sampling as well as 4 for each impactor stage) to be 3–times the standard deviation of these blanks. The blue line is an exponential fit through the values. Above a concentration of around 10 ng m⁻³, the uncertainty was mainly governed by the concentration independent volume ascertainment of around ±5%.

The original concentration data from the different impactor stages were inverted according a procedure described in Winklmayr et al. (1990). In order to assess the validity of the inversion algorithm, we first of all compared the mass balance of the original with the inverted data. The discrepancy was typically within 1% to 2% by mass, but never exceeds 5%. Next we had to consider the influence of the analytical uncertainty of the ion concentrations on the inversion procedure. To this end we run a Monte Carlo simulation, i.e. 1000 realization of the inversion for a given compound were calculated, independently varying the concentrations of each impactor stage within two SD of the experimental error (we used individual concentration dependent errors derived from the exponential fit in Fig. 2). From these 1000 realizations, the mean size distribution and the confidence intervals were determined. The result for the two most critical cases is shown in the Supplementary Material (Fig. S1). For clarity, we refrain below from displaying confidence intervals in the presented impactor data.

Comparing ionic concentrations derived from low volume and impactor sampling (including results from both campaigns covering the same sampling period, i.e. 11 out of 12 impactor samples), we found that concentration ratios impactor versus low volume sampling was 0.64±0.15 for MS$^-$, 0.79±17 for nss–SO$_4^{2-}$, 1.1±0.2 for Na$^+$, 0.96±0.4 for Cl$^-$, and 0.72±0.5 for NH$_4^+$. The somewhat low sampling efficiency of the impactor for MS$^-$ may be due to partial re–evaporation from the tedlar foils (Teinilä et al., 2014). Finally, denuder experiments conducted by Piel et al. (2006) demonstrated that spurious post–sampling reactions like mobilization of volatile acids could generally be regarded as negligible for Teflon – nylon filter combinations.

### 2.2.3 Meteorological measurements and backward trajectories

Basic meteorological parameters, as there are wind speed (U) and wind direction (D), temperature (T), barometric pressure (P), and relative humidity (rh), were measured in 60 s resolution by a compact automatic weather station (Vaisala Weather Transmitter WXT520). The origin and pathway of the air masses advected to the measuring site was assessed by backward trajectories (HYSPLIT 4.0, Hybrid Single–Particle Lagrangian Integrated trajectory, http://www.arl.noaa.gov/documents/reports/hysplit_user_guide.pdf, last access: 7 December 2017) relying on GDAS meteorological data with a spatial resolution of 1°×1° (longitude × latitude grid). The accuracy of back trajectories is generally dependent on the availability of reliable meteorological data and their spatial coverage, which is of particular concern for remote regions like the Southern Ocean and above all Antarctica (see e.g. review by Stohl, 1998). As for coastal site Neumayer we already discussed this point (Weller et al. 2014 and 2015), but for continental Antarctica, the persisting surface inversions and katabatic winds pose additional challenges to air mass trajectory analyses. Though a detailed accuracy assessment is beyond the scope this work, we tried to validate the consistency of the used back trajectories by: (i) using the 3D wind fields of the GDAS data as well as the isentropic approximation which could be more reliable for regions with sparse meteorological input data (Harris et al., 2005); (ii) varying the starting height above ground between 10 m and 100 m; (iii) calculating trajectory ensembles whose starting points were varied by ±1° longitude and ±1° latitude each, while for the height above ground 0 m, 250 m, and 500 m were chosen, respectively. The impact of these different initial conditions on our conclusions will be appraised in the Discussion section. Finally weather charts generated by the Antarctic Mesoscale Prediction System (AMPS) were used to assess the general weather situation (Powers et al., 2003; http://www2.mmm.ucar.edu/rt/amps/information/amps_esg_data_info.html, last access: 7 December 2017)

## 3 Results

### 3.1 Meteorological conditions

Figures 3 and 4 give a detailed synopsis of the measured meteorological parameters. Regarding local meteorology, bright weather conditions, i.e. clear sky conditions with a cloud amount typically well below 2 Octans prevailed, except the period between 17 Jan. and 20 Jan. 2015 (doy 17, 12:00 to doy 20, 18:00) when the impact of a cyclone (LPS15) with a centre in the south eastern Weddell Sea reached the site (a corresponding weather chart is shown in Supplementary Material Fig. S2). Note that the cloud amounts given in Figs. 3 and 4 were assessed by everyday casual visual inspection between 04:00 and 22:00 and should be rated as subjective but reasonable estimates. In the following we will especially focus on this episodic and prominent weather situation, preluded by a sharp pressure drop of 15 hPa, associated with increasing wind velocities to around 10 m s$^{-1}$, drifting snow, overcast sky, and some snow precipitation (which was notoriously difficult to distinguish from drifting snow). Corresponding Hysplit calculations predicted only marginal local precipitation around 0.2 mm water equivalent in the evening of 19 January 2015. A second prominent high wind event occurred at 15 Jan. 2016 (LPS16, Fig. 4 and Supplementary Material Figs. S3), again induced by an intense low pressure system in the Weddell Sea region. In both cases nearby coastal

site Neumayer(70°39'S, 8°15'W) experienced severe blizzards with wind velocities up to 25 m/s. But in contrast to LPS15, the sky was then only temporarily overcast in case of LPS16 at Kohnen and no precipitation has been realized.

During the continuous clear sky conditions, katabatic winds from the northeast (following the terrain slope) around 5 m s$^{-1}$ usually with a maximum shortly after midnight and a minimum in the late afternoon were typical. Frequently, sparse clear sky precipitation (diamond dust) could be observed during night and early morning. In such cases, AMPS weather charts revealed only shallow low pressure systems offshore DML and the characteristic continental anticyclone centred some hundred km east or southeast from Kohnen, also present during LPS15 and LPS16 (Supplementary Material Fig. S2 and S3).

Due to our limited meteorological data base, a thorough characterization of the atmospheric boundary layer was not feasible. Though to this end it seems reasonable to access to a detailed investigation of the summertime atmospheric boundary layer at Kohnen by van As et al. (2006). Concerning clear sky conditions, our meteorological conditions (wind vector, T, and P) were virtually comparable to a 4–day period described therein. Hence, according to van As et al. (2006), we may assume the presence of a stable boundary layer over night with strong temperature inversion (typically around 5 K m$^{-1}$ and a thickness up to 50 m), developing to a slightly convective layer (up to 50 m thick) around noon with near–neutral stability and capped by an inversion layer (van As et al., 2006).

## 3.2 Particle concentrations and related size distributions

In Figs. 5 and 6, CP concentrations measured with both CPCs (CP$_{4nm}$ and CP$_{3nm}$ during 2015 and 2016, respectively) along with the ultrafine particle concentrations between 3 nm and 25 nm (UCP$_{3-25}$) as well as between 10 nm and 25 nm (UCP$_{10-25}$) are displayed. CP concentrations were strikingly enhanced during LPS15 (1400±700 cm$^{-3}$ compared to 250±120 cm$^{-3}$ during clear sky conditions), though on the whole clearly lower compared to coastal sites like Neumayer (Weller et al., 2011). Both UCP time series were derived from the nano–DMA 3085 and DMA 3081 data, which are conflated as contour plots in Fig. 7 and Supplementary Material Figs. S4–S6. The striking feature was the strong rise in CP and most notably UCP$_{3-25}$ concentrations during the onset of LPS15. This period was characterized by elevated UCP$_{3-25}$/CP$_{4nm}$ ratios of 0.75±0.5. After recurring weather situation, i.e. the remaining observation period in 2015 and throughout the campaign in 2016, including the stormy period LPS16, UCP/CP ratios were typically below 0.3 (more precisely: UCP$_{3-25}$/CP$_{4nm}$ and UCP$_{3-25}$/CP$_{3nm}$ stayed around 0.28±0.2, while UCP$_{10-25}$/CP$_{4nm}$ and UCP$_{10-25}$/CP$_{3nm}$ was 0.18±0.07 and 0.11±0.1, respectively, Figs. 5 and 6). Table 2 summarizes particle concentrations as well as ionic composition of the aerosol during the impact of LPS15, LPS16, and clear sky conditions. While during 2015 CPC and UCP concentrations were by nearly an order of magnitude higher compared to clear sky conditions, the effect of LPS16 was not obvious.

Expectedly, the time series of the particle number size distributions (PNSD) exhibited a corresponding distinct feature (Fig. 7): During LPS15, we observed according to Dal Maso et al. (2005) a pronounced class 1 or so–called "banana–type" NPF event. Particle growth started at doy 19 (02:50) at a modal maximum of 12 nm reaching 43 nm at doy 20 (24:00) (Fig. 7). While CP and UCP concentrations exhibited distinct breaks around noon at doy 19 (Fig. 5), steady particle growth was observed throughout. The initial nucleation particle formation rate (doy 19 between 02:50 and 17:20) in the size range 3 nm to 25 nm (J$_{3-25}$ = ΔUCP$_{3-25}$/Δt) was around 0.1 s$^{-1}$. The shape of the size distributions during this NPF event were "closed" (i.e. decreasing from the modal maxima towards the lowest size bin at 3 nm). From their temporal evolution we derived a continuous growth rate of 0.6±0.08 nm h$^{-1}$ for the whole event between doy 19 (02:50) and doy 20 (24:00). In addition, a separate calculation for the first and second part of the event (boundary: noon of doy 19) resulted in virtually idendical growth rates of 0.6±0.1 nm h$^{-1}$ for both sections. A condensation sink of typically around (2.0±0.2)×10$^{-4}$ s$^{-1}$ was calculated according to Kulmala et al. (2001) from SMPS data when operated with DMA 3081, covering the extended size range between 10 nm and 420 nm.

In addition, we observed enhanced UCP concentrations between 10 nm and 25 nm during doy 18 in the morning hours, just before the actual NPF, and in the evening of doy 19 (numbered 2, 4, and 5 in Fig. 7). Apart from that, discernible natural nucleation bursts occurred around noon of doy 17 and 18 (numbered 1 and 3 in Fig. 7). All these transient UCP maxima did not show any detectable particle growth. The nucleation bursts were characterized by increasing particle concentrations from slightly above 5 nm downward towards the lower instrumental size limit ("open" distribution), indicating local nucleation. Table 3 provides a summary of the respecting particle formation rates $J_{3-25}$ and the range of the observed particle diameter $D_p$ for these events. In contrast, local contamination provoked strong particle bursts which typically showed spiky and strongly enhanced particle concentrations ($UCP_{3-25}$ concentrations $>2500$ cm$^{-3}$, $J_{3-25}$ typically $>10$ s$^{-1}$) within a wide particle size range as marked with white frames in Supplementary Material Fig. S4.

For the rest of the measuring period in 2015, PNSD were prima facie monotonous with a mode maximum around 60 nm (Fig. 8). Notably, an almost persistent Aitken mode around 34 nm was present, also obvious in the mean PNSD derived from DMA 3081 data between doy 28 and doy 33 (Supplementary Material, Fig. S5). In contrast, merely an accumulation mode could be identified in the corresponding mean PNSD covering the period from doy 23 to the end of the campaign in 2016 (Fig. 8). In the first part of the latter campaign (doy 12 through doy 22, measured with the nano–DMA 3085), an additional modal maxima between 10 nm and 30 nm sporadically appeared (Supplementary Material, Fig. S6).

### 3.3 Ionic composition of bulk– and size segregated aerosol

Though super–µm particles were not captured by the SMPS data, PNSD appeared clearly governed by sub–µm aerosol (Fig. 8), indicating a crucial role of nss–$SO_4^{2-}$ in the chemical composition of the aerosol. Indeed this was supported by the results from bulk and size segregated aerosol samplings. Throughout both campaigns, about (75±6)% of the aerosol mass consisted of biogenic sulfur aerosol (nss–$SO_4^{2-}$ and MS$^-$), leaving only about (9±5)% for sea salt aerosol, while the highly variable $NH_4^+$ portion contributed to about (6±4)%. The amount of sea salt aerosol was calculated from the measured Na$^+$ concentrations referring to standard mean ocean water composition with a Na$^+$/sea salt mass ratio of 0.306 (Holland, 1993). Note, that due to analytical problems we assumed a $NO_3^-$ portion of 7.4% in the above mass balance estimate, according to results from three former summer campaigns at Kohnen (Piel et al., 2006). Ion balance considerations revealed that the sampled aerosol was constantly acidic i.e. (0.7±0.3) neq m$^{-3}$ H$^+$ corresponding to (40±15)% H$^+$ in 2015 and (0.8±0.6) neq m$^{-3}$ H$^+$ or (25±16)% H$^+$ equivalent contingent in 2016. Note that these figures are nota bene a lower limit because ignoring here any HNO$_3$ part due to given analytical problems. Although Cl$^-$/Na$^+$ mass ratios were highly variable (1.0±0.9 in 2015 and 0.7±0.3 in 2016), a significant Cl$^-$ depletion relative to the seawater composition (Cl$^-$/Na$^+$ = 1.8) was evident, indicating HCl mobilization by acids like HNO$_3$ or H$_2$SO$_4$.

Generally ion concentrations (except $NH_4^+$) were considerably higher during LPS 15 and LPS16 compared to clear sky conditions (Table 2). The time series of nss–$SO_4^{2-}$ and MS$^-$ as well as ß$_{MS}$ derived from low volume sampling are shown in Figs. 9 and 10, while the results for Na$^+$ and $NH_4^+$ are presented in the Supplementary Material, Fig. S7. Striking features were extremely high biogenic sulfur concentrations during LPS15 and shortly after (doy 19 through doy 21, Fig. 9) in combination with notable Na$^+$ concentrations around 53 ng m$^{-3}$ (Supplementary Material Fig. S7a). Interestingly, first MS$^-$ concentrations hit its peak (190 ng m$^{-3}$) at doy 19/20, while nss–$SO_4^{2-}$ peak (137 ng m$^{-3}$) succeeded with a delay of one day. Considering the merely daily resolution of the low volume sampling procedure (during this event low volume sampling started and ended at around 16:00), enhanced biogenic sulfur concentrations seemed to arise in the final stage of LPS15. In addition, ß$_{MS}$ was considerably higher compared to the rest of the season (1.3±0.9 versus 0.18±0.03) and throughout the observation period in 2016 (0.23±0.13). Thereby ß$_{MS}$ during clear sky conditions was intermediate between those encountered in coastal regions (Legrand and Pasteur, 1998) and at Concordia (close to 0.1 from November to April, Legrand et al. (2017a)). Concerning

LPS16 on the other hand, notable marine aerosol concentrations (biogenic sulfur and $Na^+$, Fig. 10 and Supplementary Material Fig. S7b) were observed, but now clearly in the aftermath of the stormy period.

Figures 11 and 12 show the size segregated composition of the aerosol, derived from impactor samples taken 2015 and 2016, respectively. Note that in case of 2016 we could not assign particular impactor results to LPS16 due to the short duration of this event compared to the sampling period. In Fig. 11, percentage entries denote the portion of the corresponding ion mass in the size range >1 µm (super–µm or coarse mode), calculated from the inverted profiles. Generally spoken and consistent with comparable measurements conducted at Concordia (Legrand et al., 2017a), a significantly larger part of $MS^-$ compared to nss–$SO_4^{2-}$ resided in the coarse mode resulting in higher $ß_{MS}$ ratios there (range: 0.4 up to 2.3 during LPS15 when MSA reached 190 ng m$^{-3}$, Fig. 13). Concerning sea salt aerosol, nearly half of the $Na^+$ mass was present as sub–µm aerosol (Fig. 11). On the whole, ion balance considerations (again inevitably neglecting $HNO_3$) indicated acidic sub–µm aerosol while in super–µm samples nss–$SO_4^{2-}$ and $MS^-$ was roughly counterbalanced by $Na^+$ and $NH_4^+$, in agreement with previous results from Concordia during summer (Becagli et al., 2012, Legrand et al., 2017a). However, ion balance results appeared not as clear–cut as found in the bulk low volume Teflon – nylon filter combination samples: For sub–µm aerosol the mean excess anionic portion was merely 0.074 neq m$^{-3}$ in 2015 and 0.03 neq m$^{-3}$ in 2016, while for super–µm aerosol we found a mean excess cation part of 0.015 neq m$^{-3}$ (2015) and 0.084 neq m$^{-3}$ (2016). There are two main plausible reasons for this discrepancy: (i) a more pronounced loss of acidic gases from the impactor foils and (ii) an efficient sampling of gaseous HCl and $HNO_3$ on nylon filters (Piel 2004 and Piel et al., 2006). Again, during LPS15, respecting impactor results (covered mainly by the first and partly by the second impactor sampling period) exhibited some conspicuous features: Particle mass size distributions (PMSD) for $MS^-$ appeared somewhat broader and a considerably greater part of nss–$SO_4^{2-}$ as well $NH_4^+$ resided in the super–µm mode (Fig. 11). Regarding the 2016 campaign, impactor results (Figure 12) were apparently comparable to the 2015 campaign for prevailing clear sky conditions conditions (covered by impactor samples 3–5, Fig. 11).

## 4 Discussion

### 4.1 Case study LPS15: Cyclone induced marine air advection and NPF

#### 4.1.1 Bulk and size segregated chemical composition

In Dronning Maud Land cyclone driven marine air intrusions are infrequent, sporadic events that are often associated with high–precipitation rates (Birnbaum et al., 2006; Schlosser et al., 2010; Welker et al., 2014; Kurita et al., 2016). Such a cyclone induced advection of marine boundary layer air masses towards the Antarctic Plateau had essentially coined the physical and chemical properties of the aerosol on–site, most noticeable by maximum of biogenic sulfur concentrations and the occurrence of a NPF event. During a previous similar general weather situation at Kohnen (10 and 11 January 2002, LPS02), Piel et al. (2006) reported on even strikingly higher nss–$SO_4^{2-}$ (1100 ng m$^{-3}$) and $MS^-$ (350 ng m$^{-3}$) maxima, though about 48 h after passing of the frontal system and transition from marine to continental air mass origin. In contrast to the LPS15, LPS02 came along with heavy snowfall (Birnbaum et al., 2006). Another peculiarity of our recent observations was a preceding and well defined $MS^-$ peak pursued by a distinct nss–$SO_4^{2-}$ maximum (Fig. 9), compared to their simultaneous emergence in the aftermath of LPS02 described in Piel et al. (2006). Given that $MS^-$ should have been primarily formed by heterogeneous liquid phase chemistry prevalent in the marine boundary layer (Legrand et al., 2001; Bardouki et al., 2002; Hoffmann et al., 2016), the segregated $MS^-$ peak indicated a temporary and efficient advection of such air masses. In addition a striking peak of the sea–salt tracer $Na^+$ appeared along with the $MS^-$ maximum (Supplementary Material, S7), emphasizing the impact of marine boundary layer air masses during this part of LPS15. Note that $Na^+$ remained at typical mean concentrations around 7 ng m$^{-3}$ during the former biogenic sufur maximum between 14 and 15 January 2002 (Piel, 2004).

Five–day back trajectories confirmed these conclusions (Fig. 14, starting point 100 m above Kohnen): Air masses during LPS15 were generally marine 2 to 3 days before arrival at Kohnen. Especially trajectories representing doy 19 (NPF event, reddish and yellow traces in Fig. 14a) spent several hours within the marine boundary layer close to the East Antarctic coast before arrival at Kohnen, following then largely the contour lines of the local topography. All trajectories started under cyclonic curvature, finally approaching Kohnen in an anticyclonic bow from northerly directions (Fig. 14). In order to estimate the reliability of this trajectory based finding, we repeated the calculations with an initial height of 10 m above Kohnen, calculated trajectory ensembles, and in an extra attempt using the isentropic approach instead of the 3D wind field from GDAS data (Supplementary Material Figs. S8–S10). Though appreciable differences regarding the geographic location of the corresponding source regions and trajectory course were obvious, in the end the basic aforementioned implications appeared consistent. In contrast to this finding, air masses originated and stayed within continental Antarctica during the biogenic sulfur peaks observed during 14 and 15 January 2002 (Piel et al., 2006), which could be affirmed by a re–analysis now with HYSPLIT trajectories based on NCEP meteorological data (Supplementary Material Fig. S11). Similar air mass trajectories (not shown) were observed during and in the aftermath of a short stormy period LPS16, which was again characterized by enhanced $Na^+$ and biogenic sulfur loadings (Fig. 10 and Supplementary Material Fig. S7b).

Regarding the chemical composition of the aerosol during the final stage of LPS15, subsequent increasing $nss–SO_4^{2-}$ along with declining $MS^-$ concentrations indicated a minor importance of liquid phase chemistry (Hoffmann et al., 2016). We may speculate that now intrusions of marine boundary layer into the so–called buffer layer were responsible for efficient advection of gaseous DMS photo oxidation products like $SO_2$ and DMSO (Davis et al., 1998; Russel et al., 1998). Russel et al. (1998) assumed that the buffer layer typically extends from the turbulent marine boundary layer (400 m to 700 m) up to a capping inversion (1400 m to 1900 m). While transported to continental Antarctica, gas phase photo oxidation processes should have dominated, leading finally to a preferred formation of $H_2SO_4$ at the expense of MSA (Preunkert et al., 2008). Anyway, this plausible but subtle transport route could not be unequivocally deduced from respecting backward trajectory analyses, because in the case at hand, the presence and the extent of a buffer layer could not be ascertained from available meteorological data.

Another conspicuous point was the comparatively large part of $MS^-$ and most notably $NH_4^+$ found in coarse mode aerosol during LPS15 (11.5% and 39%, respectively, Fig. 11). Apart from post sampling reactions on the first two impactor stages, which could not be entirely excluded, preceding heterogeneous processes in the atmosphere i.e. acidification of sea salt particle surfaces and subsequent chemisorption of (basic) $NH_3$ should be a more realistic explanation. Though $MS^-$ added up to merely 0.03 nmol m$^{-3}$ compared to 0.072 nmol m$^{-3}$ $NH_4^+$ in coarse aerosol during LPS15, other acidic gases (probably $HNO_3$, but according to our data only to a minor part $H_2SO_4$) reacted with coarse mode sea salt particles and scavenged gaseous $NH_3$. Previous size segregated aerosol sampling at Concordia revealed that coarse mode ammonium was primarily present as sulfate or methane sulfonate salts, while nitrate was of minor importance (Becagli et al., 2012). During the Japanese–Swedish joint Antarctic expedition (JASE) traverse, on the other hand, Hara et al. (2014) reported on $H_2SO_4$ and MSA modified sea salt particles close to the coast while farther inland most probably the reaction with $HNO_3$ dominated. According to recent year–round investigations at Concordia, however, it appears that the competing role of $HNO_3$ vs. acidic sulfur aerosol is more complicated: Regarding coarse mode sea salt particles, only during mid– and late summer, sulfuric aerosol and not $HNO_3$ become the preferred acidic reactant (Legrand et al., 2017b).

In conclusion it is worthwhile considering results from Neumayer. This coastal site was governed by the same low pressure system (LPS15) provoking there a blizzard around 19 January 2015. From there, daily low volume sampling (Teflon–nylon filter combination), CP concentration (measured with a CPC 3022A, TSI, $D_{p(50\%)}$ = 7 nm), and meteorological data were available. Again, biogenic sulfur (particularly $MS^-$), $Na^+$ and CP concentrations showed distinctive maxima though about one day after the LPS15 (Supplementary Material Fig. S12), similar to the previous situation described by Piel et al. (2006).

Consequently we can assume that this characteristic weather situation transported marine aerosol throughout DML. But in contrast to Kohnen the impact of LPS15 on the delayed particle concentration and ionic composition maxima at Neumayer was less pronounced. Pant et al. (2010) concluded from previous particle concentration and size distribution data measured at coastal Maitri that during the impact of cyclones, coarse mode sea salt aerosol increased by an order of magnitude compared to calm weather conditions, similar to our results during LPS15 (Table 2).

### 4.1.2 NPF, particle growth, and size segregated chemical composition

Obviously the most striking feature during LPS15 was a distinctive NPF event. The closed shape of PNSD, starting with a modal maximum not lesser than 12 nm, implied that the actual particle nucleation event should have occurred upwind of Kohnen. Assuming a constant growth rate of 0.6 nm h$^{-1}$ throughout advection to the measuring site, particle nucleation happened thus about 20 hours before, i.e. according to backward trajectories roughly 700 km away from Kohnen at 73°S, 19°E and 160 m above ground (all virtually independent from the choice of the starting height). Note, that the observed particle growth was confined to the nucleation mode, and consequently did not reach a size range potentially relevant for acting as CCN. As described in Weller et al. (2015), NPF events at Neumayer generally showed almost comparable growth rates. Therein we used a simple estimate, based on Nieminen et al. (2010) and Yli–Juuti et al. (2011) that demonstrated the need of other condensable vapours than sulfuric acid to sustain the observed particle growth (Weller et al., 2015). We employed now the same approach in connection with the NPF event at Kohnen (i.e. eq. (3) in Weller et al., 2015, using T=250 K), resulting in $1.4 \times 10^7$ molec cm$^{-3}$ gaseous H$_2$SO$_4$ necessary for the determined growth rate of 0.6 nm h$^{-1}$. Again, this appeared at least an order of magnitude too high compared to respecting values yet published (e.g. Mauldin III et al., 2004) emphasizing the importance of other, yet unknown condensable vapours, most probably low volatile organic compounds (Metzger et al., 2010, Tröstl et al., 2016). Interestingly, Kyrö et al. (2013) identified biogenic emissions by nearby melting ponds as a potential source for condensable vapour. The surroundings of Kohnen, however, are completely ice covered throughout as typical for the Antarctic Plateau region. The nearest rocky outcrops are more than 200 km away.

In this context it is interesting to compare our results with previous PNSD measurements conducted at South Pole during two summer campaigns (Park et al., 2004) and year–round observations at Concordia (Järvinen et al., 2013). Results from South Pole showed similar PNSD with mean a $D_p$ between 60 nm and 98 nm (Park et al., 2004, Table 4 therein), while PNSD at Concordia were appreciably lower with $D_p$ = 39 nm (Järvinen et al., 2013). Järvinen et al. (2013) could detect NPF throughout the year under condensation sink values of around $1.8 \times 10^{-4}$ s$^{-1}$, comparable to the condensation sink range found at Kohnen. At Concordia, NPF events with determinable growth rate were not connected with low pressures systems but essentially observed under air mass advection from the upper atmosphere (Järvinen et al., 2013). This fact indicates that also at Kohnen, NPF may potentially occur during clear sky conditions, but could not be detected within our admittedly limited observation period. Furthermore, growth rates determined so far at coastal sites (Virkkula et al., 2007; Asmi et al., 2010; Kyrö et al., 2013; Weller et al., 2015) appeared comparable to GR reported from continental Antarctica (Järvinen et al., 2013; Chen et al., 2017) and were within a similar broad range between some 0.2 nm h$^{-1}$ up to 8.8 nm h$^{-1}$.

Pant et al. (2010) presented a detailed analysis about the impact of passing cyclones on particle size distributions at coastal Maitri, These authors observed bimodal PNSD with a coarse mode maxima around 2 µm and a broad Aitken mode between 0.04 µm and 0.1 µm when a storm approached the site. Occasionally NPF occurred just after the passage of a cyclone associated with particle growth rates between 0.2 and 0.6 nm h$^{-1}$. From meteorological data these authors conclude that the observed NPF events were linked with mixing of marine and continental during subsidence of free tropospheric air after the storm (Pant et al., 2010).

### 4.2 The standard case: Clear sky conditions and aged aerosol

 clear sky conditions largely prevailed during our recent, but also during previous summer campaigns (Piel et al., 2006). The present observations at Kohnen showed that throughout this characteristic synoptic situation, appreciably lower particle number concentrations restricted within the accumulation mode were typical. Concerning the ionic composition, especially $MS^-$ and $Na^+$ concentrations were considerably lower compared to their maxima accompanied with the impact of LPS15 (Fig. 9 and Supplementary Material Fig. S7a), in contrast to virtually constant, though highly variable mean $NH_4^+$ concentrations (5.2±2.6 ng m$^{-3}$ in 2015 and 17.9±6 ng m$^{-3}$ in 2016). We have no explanation at hand for the about threefold higher mean $NH_4^+$ concentrations in 2016, but results presented in Piel et al. (2006) also showed large annual fluctuations. We finally calculated a mass ratio of $NH_4^+$ to $nssSO_4$ of 0.06 for 2015 and 0.15 for 2016. At least the former value was in the range of what was concurrently observed at Neumayer in January 2015. Most probably atmospheric $NH_3/NH_4^+$ ratios were correspondingly variable depending on the availability of acidic trace gases (note, that $NH_3$ could not be captured by the used sampling methods).

Turning towards biogenic sulfur aerosol, $ß_{MS}$ ratios decreased to values around 0.2 under clear sky conditions, as typical for continental Antarctica (Piel et al., 2006, Weller and Wagenbach 2007, Preunkert et al., 2008, Becagli et al., 2012), while during LPS15, $ß_{MS}$ was more comparable to coastal sites like Neumayer and Dumont D'Urville (Legrand and Pasteur, 1998).

The air mass history during clear sky conditions was assessed by respecting composite backward trajectory calculations and is summarized in Fig. 15. Obviously even 10 days before arrival at Kohnen with a characteristic anticyclonic curvature, trajectory origins remained principally inside the Antarctic continent and thus remote from marine source regions. Varying initial start height did not essentially change this general feature, but employing the isentropic instead of the 3D approach showed an increased relevance of marine source regions (see Supplementary Material Fig. S13). Upon extending the trajectory travel time to 20 days, the origin of the air masses became eventually marine (Fig. 16), covering now a large part of the Southern Ocean, except the western part of the Weddell Sea and the Bellingshausen Sea. The relevance of air mass transport via the free troposphere was difficult to assess, mainly due to the generally highly variable and poorly characterized depth of the marine boundary layer and especially the vertical extent of the atmospheric boundary layer over Antarctica. According to Russell et al. (1998), the border between the marine boundary layer and the free troposphere varies typically between 1400 m and 1900 m while on the Antarctic Plateau only a shallow atmospheric boundary layer of no more than a few hundred meters is typical (van As et al., 2006). Anyway, an inspection of the calculated trajectories revealed that they typically originated within the marine boundary layer (i.e. mainly below 1500 m) and essentially stayed below 500 m above ground across continental Antarctica.

Surprisingly, while the amount of sea salt aerosol was highly variable, the super–µm mode fraction of sea salt aerosol remained constant at around 50% during rapid and efficient marine boundary layer air mass advection under LPS15 as well as during long range transport under clear sky conditions (Fig. 11 and 12). Since our sea salt mass size distribution appeared also similar to that typically observed at Concordia (Jourdain et al., 2008; Legrand et al., 2017b), we infer that transport of coarse mode sea salt particles to continental Antarctica was generally inefficient, irrespective the general weather situation and transport time. Finally, the fact that observed $ß_{MS}$ ratios were in general higher in the coarse mode (supported by measurements in coastal Antarctica by Teinilä et al. (2000) and Rankin and Wolff (2003)), implied a preferential loss of $MS^-$ during transport towards the Antarctic plateau region. In contrast, recent results from Concordia give strong evidence for a preferential photochemical depletion of MSA in the atmosphere above the plateau during austral summer (Legrand et al., 2017a) but less indication for fractional loss en route. Nevertheless, both processes, fractionation as well photochemistry could be potential explanations for

typically lower $\beta_{MS}$ ratios commonly found in continental Antarctica compared to the coastal regions (Piel et al., 2006, Weller and Wagenbach 2007, Preunkert et al., 2008).

## 5 Conclusions

We measured aerosol size distributions and conducted bulk as well as size segregated aerosol sampling during two summer campaigns at the continental Antarctic station Kohnen. This extended approach allowed a detailed synopsis of the physical and chemical properties of summer aerosol in this region. For the first time notably the impact of passing cyclones on aerosol advection into the Antarctc Plateau region was examined. Based on these, admittedly still limited investigations, we may conclude that during austral summer, transport of marine aerosol to Kohnen in particular, and to continental DML in general, was mediated by two different synoptic situations: (i) impact of low pressure systems in the western part of the South Atlantic, associated with temporarily exceptional marine aerosol concentrations, (ii) endured long range transport providing a background aerosol level during clear sky conditions over DML. In the present study, a distinct low pressure event (LPS15) was additionally associated with NPF. Under prevailing clear sky conditions, on the other hand, aged and less aerosol (by mass and number concentration) entered DML in air masses which were typically continental for about 10 days before. We tentatively infer that our recent observation, i.e. NPF and peaking marine aerosol concentrations during LPS15 could be of fortuitous occurrence since unexpectedly just then trace compounds seemed hardly be depleted by precipitation. In contrast during the mentioned striking blizzard in 2002 (LPS02) reported by Piel et al. (2006), biogenic sulfur concentrations stayed first quite low but peaked about 48 hours in the aftermath of the storm. Though we refer on two recent and three previous studies, just three LPS and one pronounced NPF event occurred that could be analysed in detail, emphasizing their sporadic nature. Hence, a worthwhile confirmation of our conclusions would clearly require similar investigations at this site. Such effort is important to better understand the role of biogenic aerosol in general and in particular the impact of NPF events on regional climate forcing.

Anyhow, though efficient transport of biogenic sulfur (and also sea salt) aerosol to continental DML may be associated with cyclonic activity in the South Atlantic, in the long run the crucial transport pathway of marine aerosol during austral summer should be long–range transport under typical clear sky conditions. In particular for biogenic sulfur, showing a pronounced summer maximum (Weller and Wagenbach 2007), we suppose that transport to DML, deposition as well as final storage in firn and glacial ice will be dominated by prevailing clear sky conditions. Thus dry deposition but to an only minor extent wet deposition (partly associated with clear sky precipitation) would be decisive. Consequently future research activities should also envisage assessing dry deposition velocities at this site, e.g. by gradient and/or eddy correlation studies (Grönlund et al., 2002; Contini et al., 2010). On the other hand, retrieving meaningful historic aerosol concentrations from ice core archives also needs a thorough consideration of snow accumulation since snow accumulation co–determine trace compound concentration in firn and ice (Fischer et al., 1998), which is evidently governed by the infrequent impact of low pressure systems (Birnbaum et al., 2006; Schlosser et al., 2010; Welker et al., 2014; Kurita et al., 2016). Finally, trajectory analyses indicated that a large part of the Southern Ocean should be considered as potential source region representative for aerosol deposition in continental DML, in contrast to coastal Neumayer where the dominance of the South Atlantic was evident (Minikin et al., 1998).

*Data availability.* Data from both campaigns reported here are available at https://doi.pangaea.de/10.1594/PANGAEA.882375 for scientific purposes. In this case, we expect collaboration with Rolf Weller (contact: rolf.weller@awi.de).

*Competing interests.* The authors declare that they have no conflict of interest.

*Acknowledgements.* The authors would especially like to thank all technicians present at Kohnen Station, namely Holger Schubert, Torsten Langenkämper and last not least Jens Köhler, whose outstanding engagement actually enabled both air chemistry campaigns at this site. We are thankful to NOAA Air Resources Laboratory for having made available the HYSPLIT trajectory calculation program as well as all used input data files. We thank Kevin Manning for providing us weather charts based on the Antarctic Mesoscale Prediction System (AMPS). Finally we appreciate the two anonymous reviewers for their helpful comments.

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

Table 1. Survey of the experimental set-up during both summer campaigns at Kohnen station

| Measured parameter | Summer campaign 2015 (between 16 Jan. 2015 and 3 Feb. 2015) | Summer campaign 2016 (between 12 Jan. 2016 and 29 Jan. 2016) |
|---|---|---|
| Particle number concentration | CNC 3775 (TSI), cut-off $D_p(50\%) = 4$ nm; 60 s resolution | CNC 3025A (TSI), cut-off $D_p(50\%) = 3$ nm; 60 s resolution |
| Particle size distribution | SMPS 3936 / WCPC 3788 (TSI); 10 min res.: 1. nano-DMA 3085 (TSI), range (nominal): 2 nm to 64 nm, measuring period: 16 Jan. to 27 Jan. 2. long DMA 3081 (TSI), range (nominal): 10 nm to 420 nm, measuring period: 27 Jan. to 2 Feb. | SMPS 3936 / WCPC 3788 (TSI); 10 min res.: 1. nano-DMA 3085 (TSI) Range (nominal): 2 nm to 64 nm Measuring period: 12 Jan. to 23 Jan. 2. long DMA 3081 (TSI), range (nominal): 10 nm to 420 nm, measuring period: 23 Jan. to 28 Jan. |
| Bulk aerosol sampling | Teflon nylon filter combination, about 24 h or 48 h sampling intervals | Teflon nylon filter combination, about 24 h sampling interval |
| Size segregated aerosol sampling | 5-stage Berner type impactor: cut-off (µm): 3.5, 1.2, 0.4, 0.12, 0.042, about 48 h or 72 h sampling intervals | 5-stage Berner type impactor: cut-off (µm): 3.5, 1.2, 0.4, 0.12, 0.042, about 48 h or 72 h sampling interval |
| Meteorology | Weather transmitter WXT520 (Vaisala), U, D, T, P, RH; 60 s resolution | Weather transmitter WXT520 (Vaisala), U, D, T, P, RH; 60 s resolution |

Table 2. Particle number concentrations and ionic composition (mean ± SD) classified into different general weather conditions (impact of low pressure system and clear sky conditions during both campaigns); note that the ionic composition during LPS16 was based on one sample hence no SD can be given.

| Measured parameter | Summer campaign 2015 | | Summer campaign 2016 | |
|---|---|---|---|---|
| | LPS15 | clear sky cond. | LPS16 | clear sky cond. |
| $CPC_{4nm}$ (cm$^{-3}$) | 1400±700 | 250±120 | n.a. | n.a. |
| $CPC_{3nm}$ (cm$^{-3}$) | n.a. | n.a. | 150±40 | 200±120 |
| $UCP_{3-25nm}$ (cm$^{-3}$) | 880±730 | 90±100 | 57±100 | 120±164 |
| $UCP_{10-25nm}$ (cm$^{-3}$) | n.a. | 46±24 | n.a. | 18±30 |
| $MS^-$ (ng m$^{-3}$) | 111±70 | 14.8±3.7 | 73.6 | 26±19 |
| $nss–SO_4^2$ (ng m$^{-3}$) | 96±35 | 81±9 | 181 | 115±24 |
| $\text{\ss}_{MS}$ ($MS^-/nss–SO_4^{2-}$) | 1.14±0.8 | 0.18±0.03 | 0.41 | 0.22±0.13 |
| $Cl^-$ (ng m$^{-3}$) | 15±11 | 2.6±2 | 7.2 | 2.6±1.3 |
| $Na^+$ (ng m$^{-3}$) | 33±20 | 2.8±1.4 | 13.6 | 4.1±1.5 |
| $Cl^-/Na^+$ | 0.45±0.3 | 1.24±1.2 | 0.53 | 0.7±0.4 |
| $NH_4^+$ (ng m$^{-3}$) | 4.0±3.8 | 5.7±2.4 | 13.2 | 18±6 |

Table 3. List of distinct enhanced UCP concentrations and natural nucleation burst events apart from the main NPF (event numbers refer to Fig. 7). The initial nucleation rate (mean ± SD) is specified together with the period of the $UCP_{3-25}$ rise and the range of the particle diameter $D_p$.

| event | Event start | Event end | $J_{3-25}$ (s$^{-1}$) | Period of $UCP_{3-25}$ rise | $D_p$ range (nm) |
|---|---|---|---|---|---|
| 1 | 17 Jan., 12:00 | 17 Jan., 15:00 | 0.05±0.02 | 12:00 to 15:00 | < 3.0 to 6.0 |
| 2 | 18 Jan., 00:00 | 18 Jan., 12:00 | 0.06±0.02 | 00:00 to 02:00 | 8.0 to 25 |
| 3 | 18 Jan., 13:00 | 18 Jan., 14:00 | 0.15±0.03 | 13:00 to 14:00 | < 3.0 to 8.0 |
| 4 | 18 Jan., 18:00 | 18 Jan., 24:00 | 0.24±0.04 | 18:00 to 21:00 | 6.0 to 40 |
| 5 | 19 Jan., 17:00 | 19 Jan., 24:00 | 0.11±0.03 | 17:00 to 19:00 | 8.0 to 12 |

**Figure captions**

Figure 1. Location of Kohnen station (Dronning Maud Land, DML) and site plan of the station surroundings.

Figure 2. Relative uncertainty $\varepsilon_{rel}(i)$ of the ion measurements derived from bulk (low volume) and size segregated (Berner impactor) sampling as a function of $\xi_i$. The circles represent the percentage error at different concentrations $\xi_i$ i.e. the measured concentration of the ion $[x_i]$ minus the corresponding $DL_i$. The blue line is an exponential fit through these values. Above a concentration of around 10 ng m$^{-3}$, the uncertainty was mainly governed by the concentration independent volume ascertainment of around 5% (red horizontal line).

Figure 3. Time series of the measured meteorological parameters in 2015 (60 s temporal resolution). The period of LPS15 is shaded in yellow. The cloud amount in Octans is denoted on the top of the second figure, based on casual visual inspection.

Figure 4. Time series of the measured meteorological parameters in 2016 (60 s temporal resolution). The stormy period LPS16 is shaded in yellow. Again, the cloud amount is denoted on the top of the second figure.

Figure 5. CP (blue diamonds), UCP$_{3–25}$ (red circles), and UCP$_{10–25}$ (purple circles) concentration time series (10 min resolution) during the campaign 2015. UCP$_{3–25}$ and UCP$_{10–25}$ values were derived from the nano DMA 3085 and DMA 3081 data, respectively. The period of LPS15 is shaded in yellow.

Figure 6. CP (blue diamonds), UCP$_{3–25}$ (red circles), and UCP$_{10–25}$ (purple circles) concentration time series (10 min resolution) during the campaign 2016. The period of the LPS16 is shaded in yellow.

Figure 7. Time series of the particle size distribution dN/dlogDp (cm$^{-3}$) on a logarithmic scale (color code at the top of the contour plot) including the NPF event, measured with the nano–DMA 3085. Particle growth is displayed as bold white line, derived from log–normal distribution fits through size distributions measured between 19 January 02:55 and 20 January 24:00. The black circles represent the fitted mode mean diameters. Periods with enhanced UCP concentrations and nucleation bursts are numbered.

Figure 8. Mean PNSD during clear sky conditions measured with DMA 3081 during the campaign in 2015 (blue circles) and 2016 (turquoise circles), respectively. The red lines are lognormal fits with geometric mean diameter of 34 nm and 58 nm for the bimodal distribution observed 2015, and 63 nm for 2016, respectively.

Figure 9. Time series of the measured MS$^-$ and nss–SO$_4^{2-}$ concentrations as well as the MS$^-$/nss–SO$_4^{2-}$ mass ratio ß$_{MS}$ from bulk aerosol (low volume) sampling during the campaign 2015. The period of LPS15 is shaded in yellow.

Figure 10. Time series of the measured MS$^-$ and nss–SO$_4^{2-}$ concentrations as well as the MS$^-$/nss–SO$_4^{2-}$ mass ratio ß$_{MS}$ from bulk aerosol (low volume) sampling during the campaign 2016. The period of LPS16 is shaded in yellow.

Figure 11. Results from size segregated (Berner impactor) sampling during the campaign 2015. Bold reddish lines are the mass size distributions during LPS15. Percentage entries in the legend denote the portion of the corresponding ion mass in the size

range >1 µm (super–µm or coarse mode), calculated from the inverted profiles. The bold gray line is the median mass size distribution of the corresponding ion.

Figure 12. Results from size segregated (Berner impactor) sampling during the campaign 2016 with the following mass portion of the respective ions in the super–µm range (>1µm): $MS^-$ 14% (a), $nss–SO_4^{2-}$ 3% (b), $Na^+$ 46% (c), and $NH_4^+$ 15% (d). The bold gray line is the median mass size distribution of the corresponding ion.

Figure 13. Mean size segregated results for the $MS^-/nss–SO_4^{2-}$ mass ratio $\beta_{MS}$ determined for each of the 5 impactor stages for both seasons as well as for the event LPS15.

Figure 14. Five day backward trajectories during the NPF event, calculated with a trajectory starting height of 100 m above Kohnen at the points in time given in the legend (a). Below, the travel height above ground (local topography) is illustrated in a color coded scale (point interval 1 hour) (b).

Figure 15. Daily 10–day backward trajectories (3D approach, starting height 100 m) during clear sky conditions in 2016 (doy 12 to doy 31, N = 80). Shown is the relative (percentage) number of trajectory intersection on a given grid cell (resolution $1°×1°$).

Figure 16. Daily 20–day backward trajectories (3D approach, starting height 100 m) during clear sky conditions in 2016 (doy 12 to doy 31, N = 80). Shown is the relative (percentage) number of trajectory intersection on a given grid cell (resolution $1°×1°$).

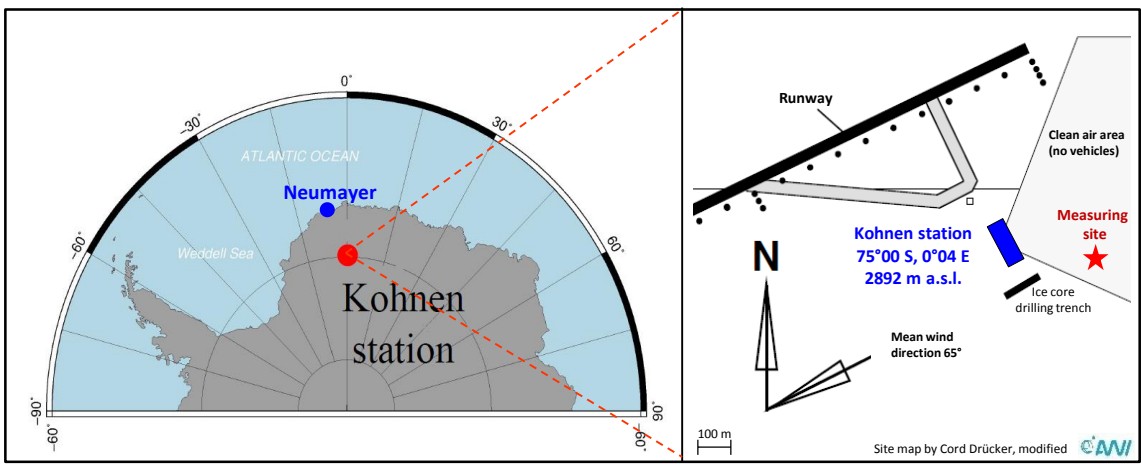

**Figure 1.** Location of Kohnen station (Dronning Maud land) and site plan of the station surroundings.

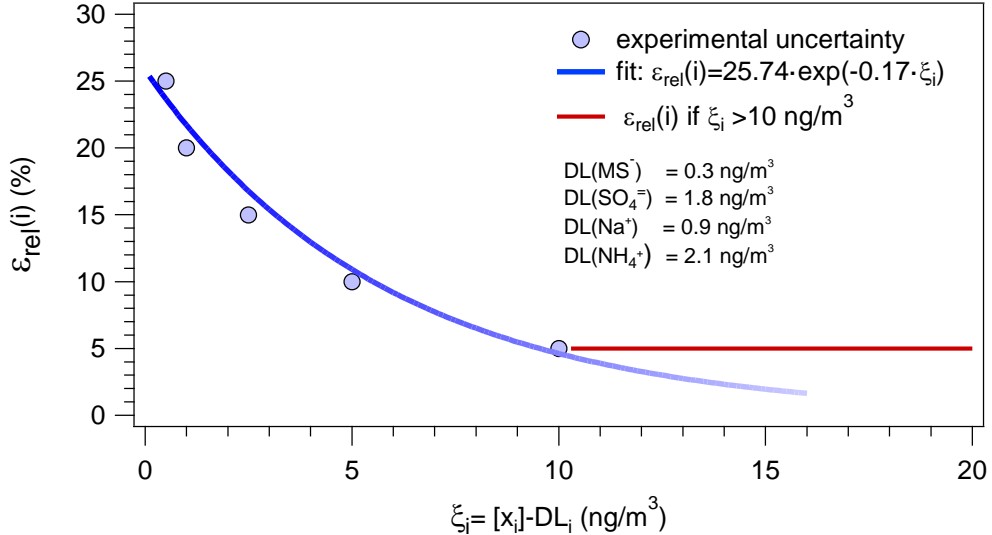

**Figure 2.** Relative uncertainty $\varepsilon_{rel}(i)$ of the ion measurements derived from bulk (low volume) and size segregated (Berner impactor) sampling as a function of $\xi_i$. The circles represent the percentage error at different concentrations $\xi_i$ i.e. the measured concentration of the ion $[x_i]$ minus the corresponding $DL_i$. The blue line is an exponential fit through these values. Above a concentration of around 10 ng m$^{-3}$, the uncertainty was mainly governed by the concentration independent volume ascertainment of around 5% (red horizontal line).

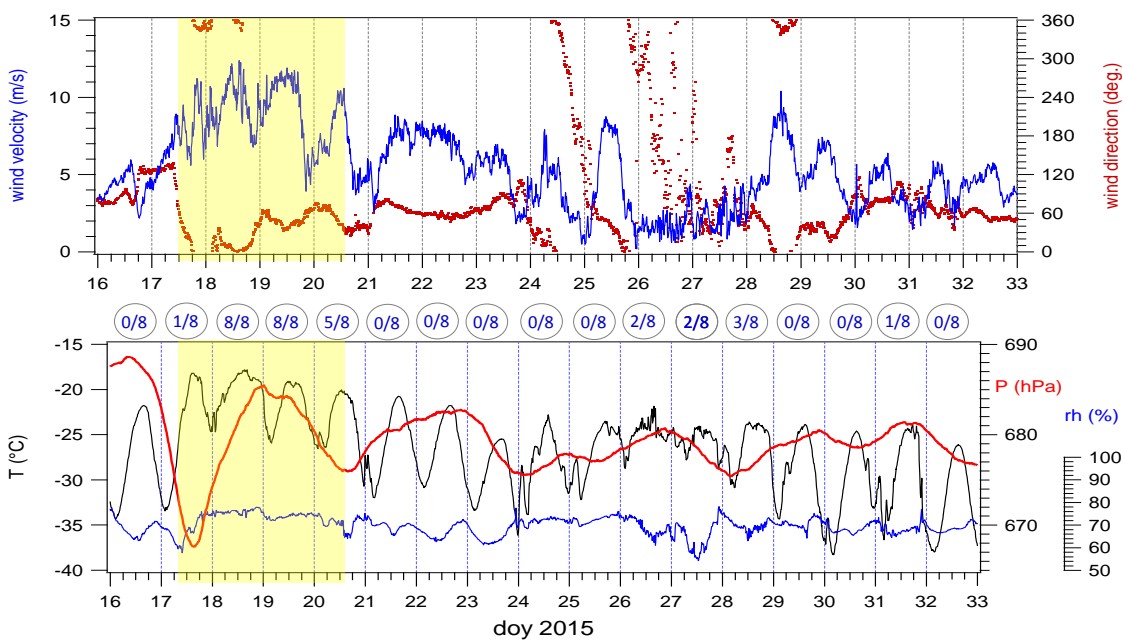

**Figure 3**. Time series of the measured meteorological parameters in 2015 (60 s temporal resolution). The period of LPS15 is shaded in yellow. The cloud amount in Octans is denoted on the top of the second figure, based on casual visual inspection.

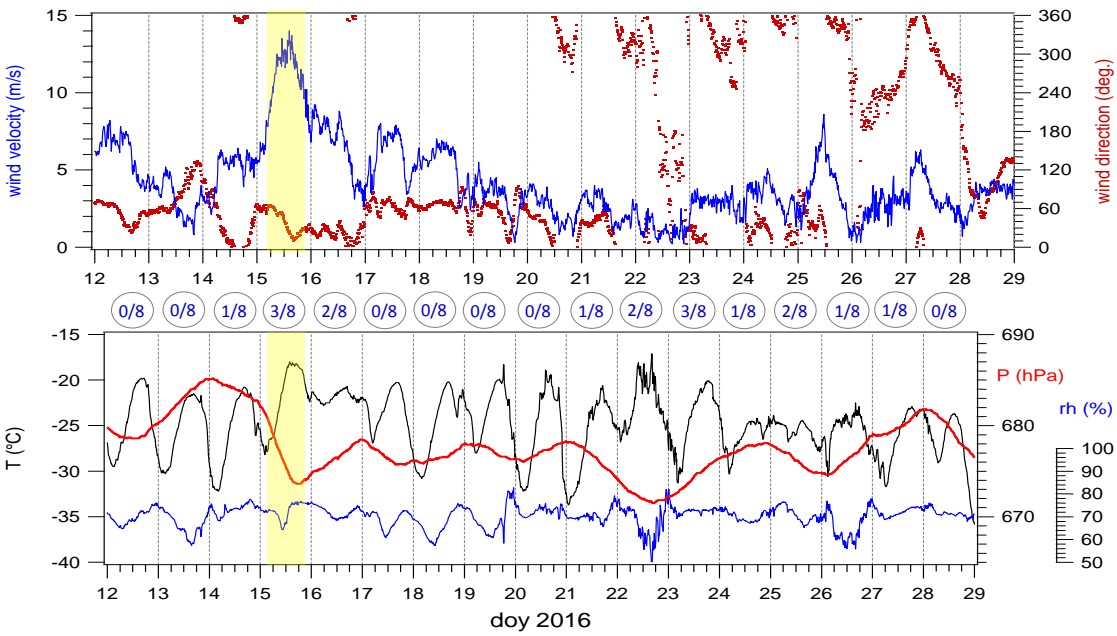

**Figure 4**. Time series of the measured meteorological parameters in 2016 (60 s temporal resolution). The stormy period LPS16 is shaded in yellow. Again, the cloud amount is denoted on the top of the second figure.

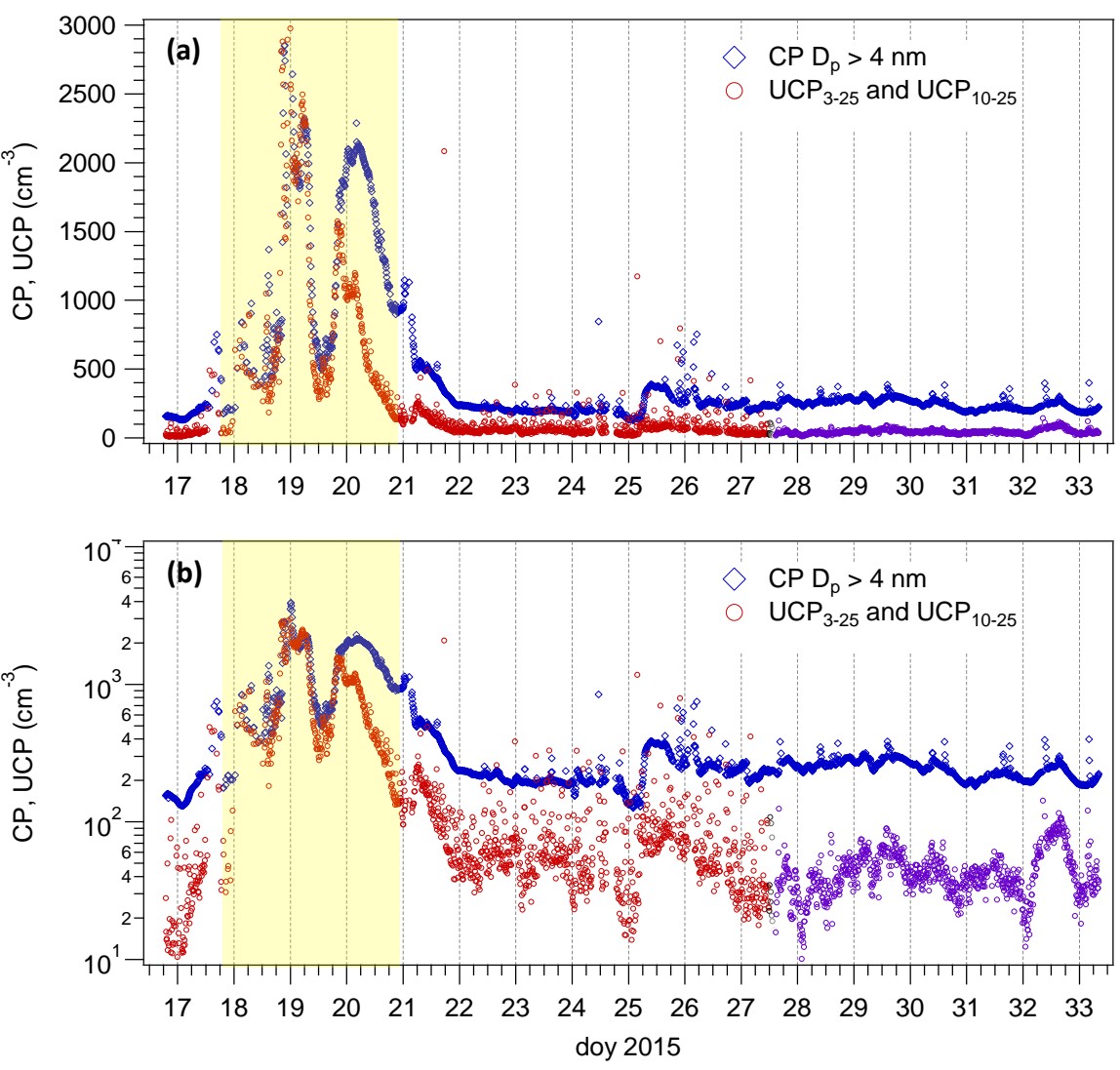

**Figure 5.** CP (blue diamonds), UCP$_{3-25}$ (red circles), and UCP$_{10-25}$ (purple circles) concentration time series (10 min resolution) during the campaign 2015. UCP$_{3-25}$ and UCP$_{10-25}$ values were derived from the nano DMA 3085 and DMA 3081 data, respectively (a). For clarity, the concentrations are additionally displayed on a logarithmic scale (b). The period of LPS15 is shaded in yellow.

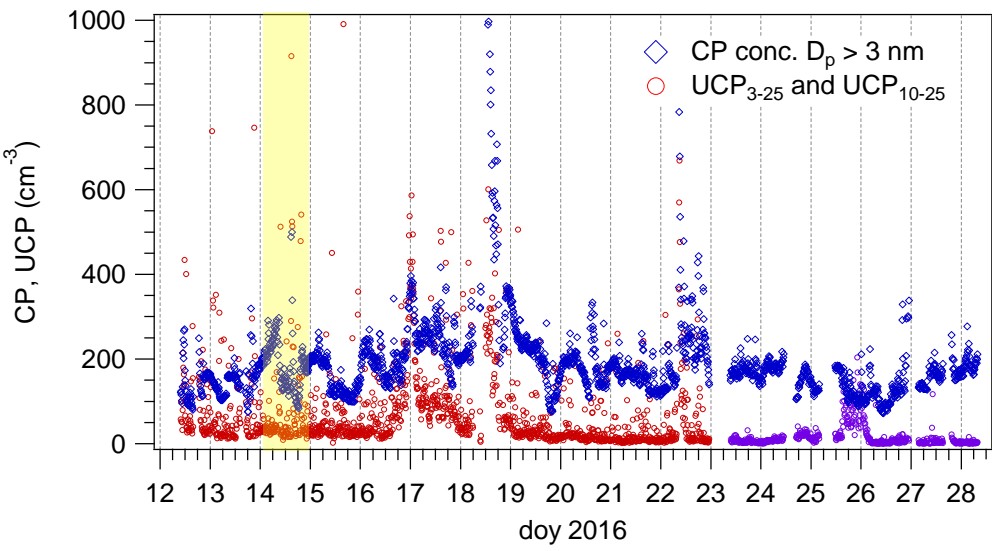

**Figure 6.** CP (blue diamonds), UCP$_{3\text{-}25}$ (red circles), and UCP$_{10\text{-}25}$ (purple circles) concentration time series (10 min resolution) during the campaign 2016. The period of LPS16 is shaded in yellow.

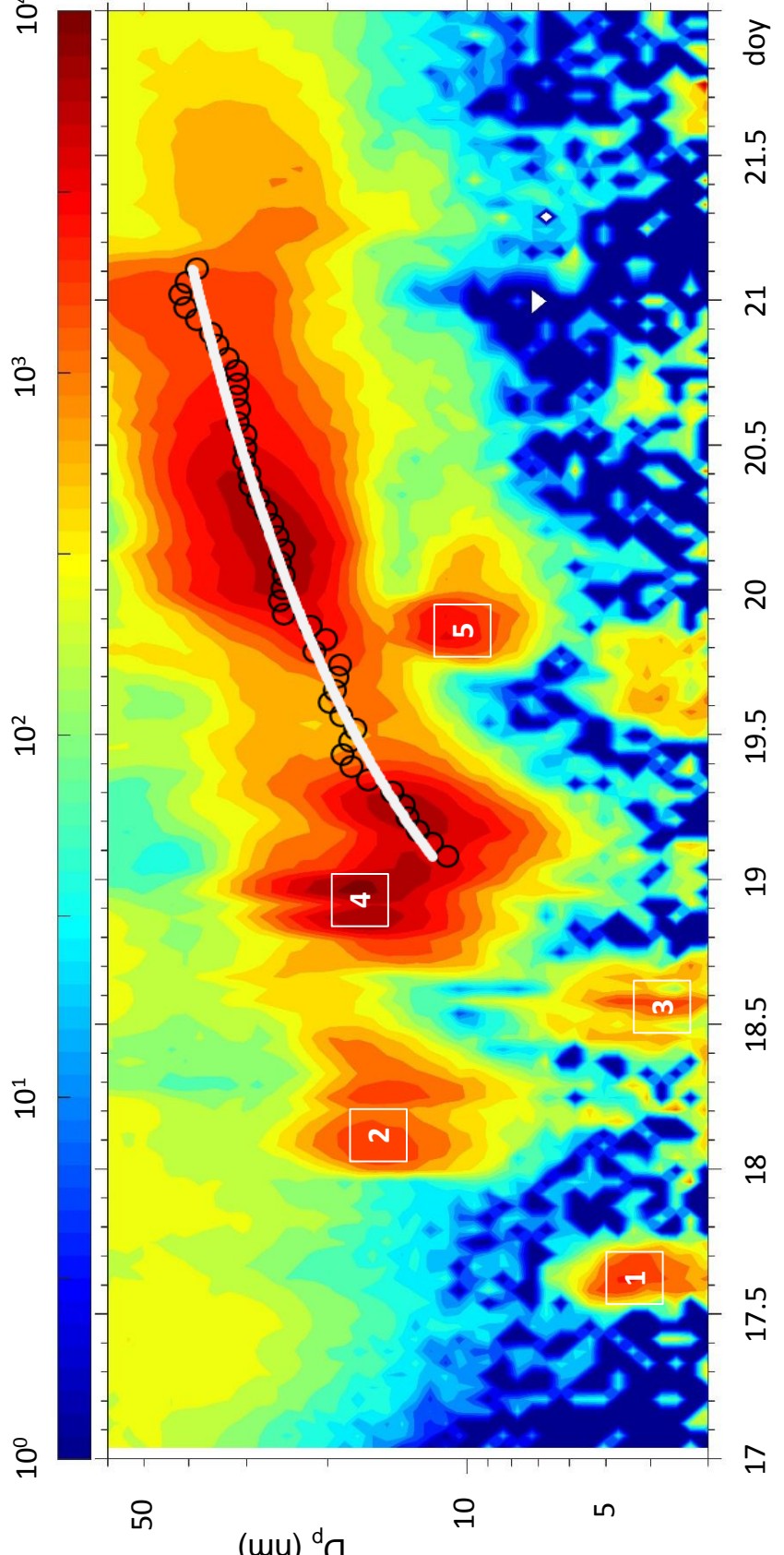

**Figure 7.** Time series of the particle size distribution dN/dlogDp (cm$^{-3}$) on a logarithmic scale (color code at the top of the contour plot) including the NPF event, measured with the nano-DMA 3085. Particle growth is displayed as bold white line, derived from log-normal distribution fits through size distributions measured between 19 January 02:55 and 20 January 24:00. The black circles represent the fitted mode mean diameters. Periods with enhanced UCP concentrations and nucleation bursts are numbered.

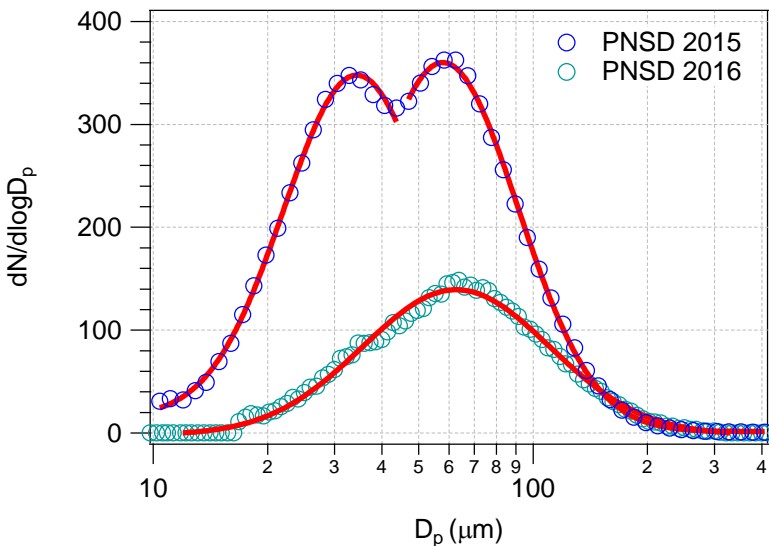

**Figure 8.** Mean PNSD during CSC measured with DMA 3081 during the campaign in 2015 (blue circles) and 2016 (turquoise circles), respectively. The red lines are lognormal fits with geometric mean diameters of 34 nm and 58 nm for the bimodal distribution observed 2015, and 63 nm for 2016, respectively.

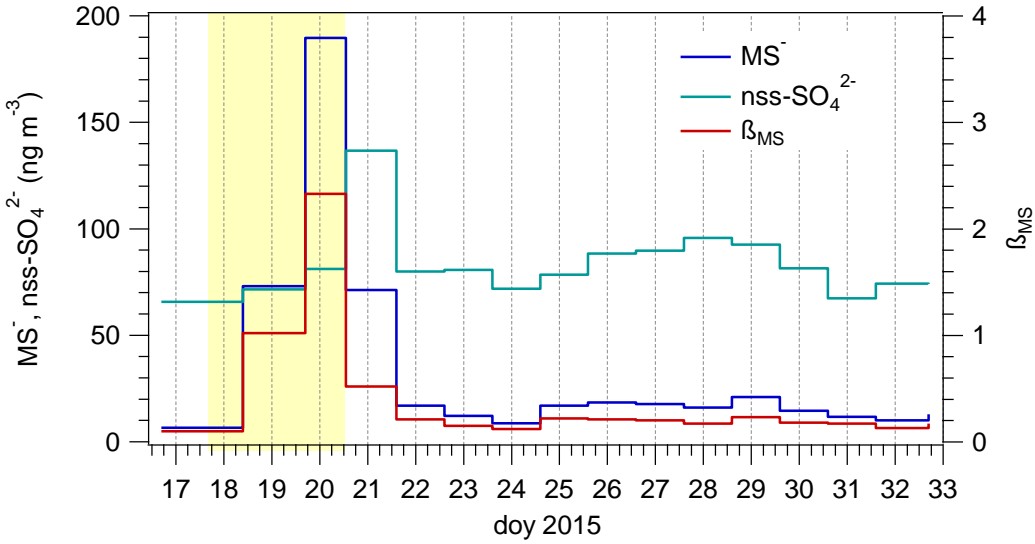

**Figure 9.** Time series of the measured $MS^-$ and $nss\text{-}SO_4^{2-}$ concentrations as well as the $MS^-/nss\text{-}SO_4^{2-}$ mass ratio $\beta_{MS}$ from bulk aerosol (low volume) sampling during the campaign 2015. The period of LPS15 is shaded in yellow.

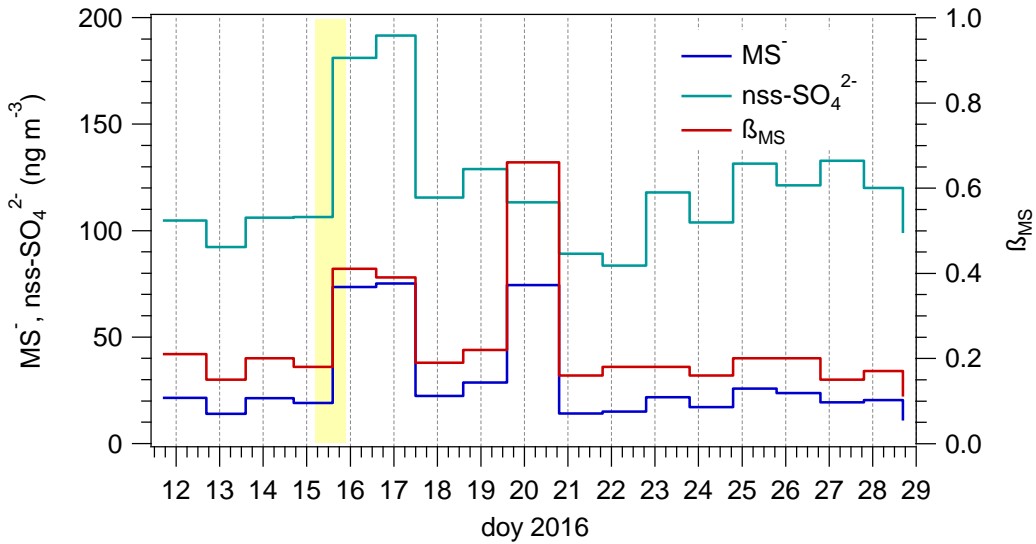

**Figure 10.** Time series of the measured $MS^-$ and $nss\text{-}SO_4^{2-}$ concentrations as well as the $MS^-/nss\text{-}SO_4^{2-}$ mass ratio $\beta_{MS}$ from bulk aerosol (low volume) sampling during the campaign 2016. The period of LPS16 is shaded in yellow.

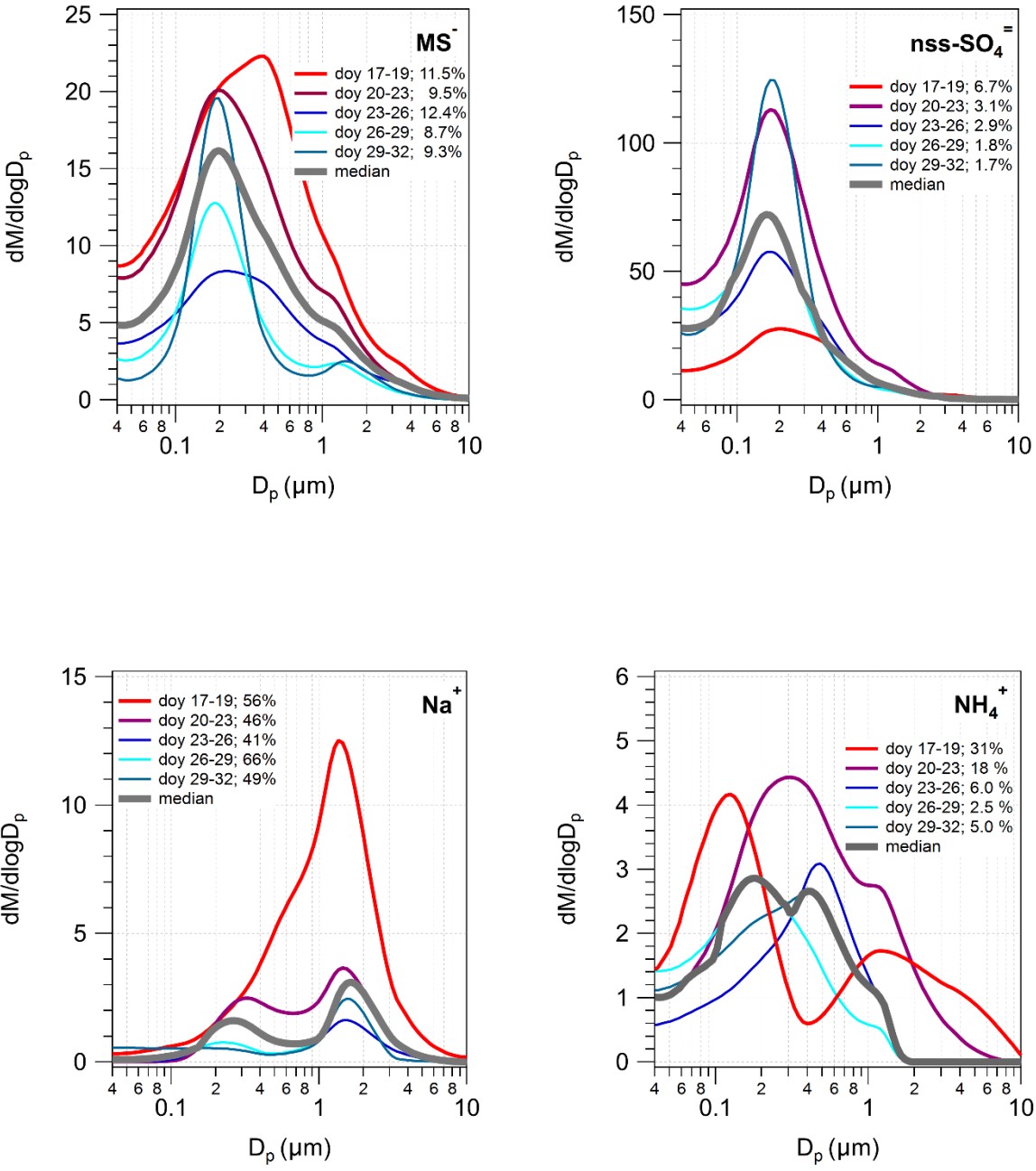

**Figure 11.** Results from size segregated (Berner impactor) sampling during the campaign 2015. Bold reddish lines are the mass size distributions during LPS15. Percentage entries in the legend denote the portion of the corresponding ion mass in the size range >1 µm (super-µm or coarse mode), calculated from the inverted profiles. The bold gray line is the median mass size distribution of the corresponding ion.

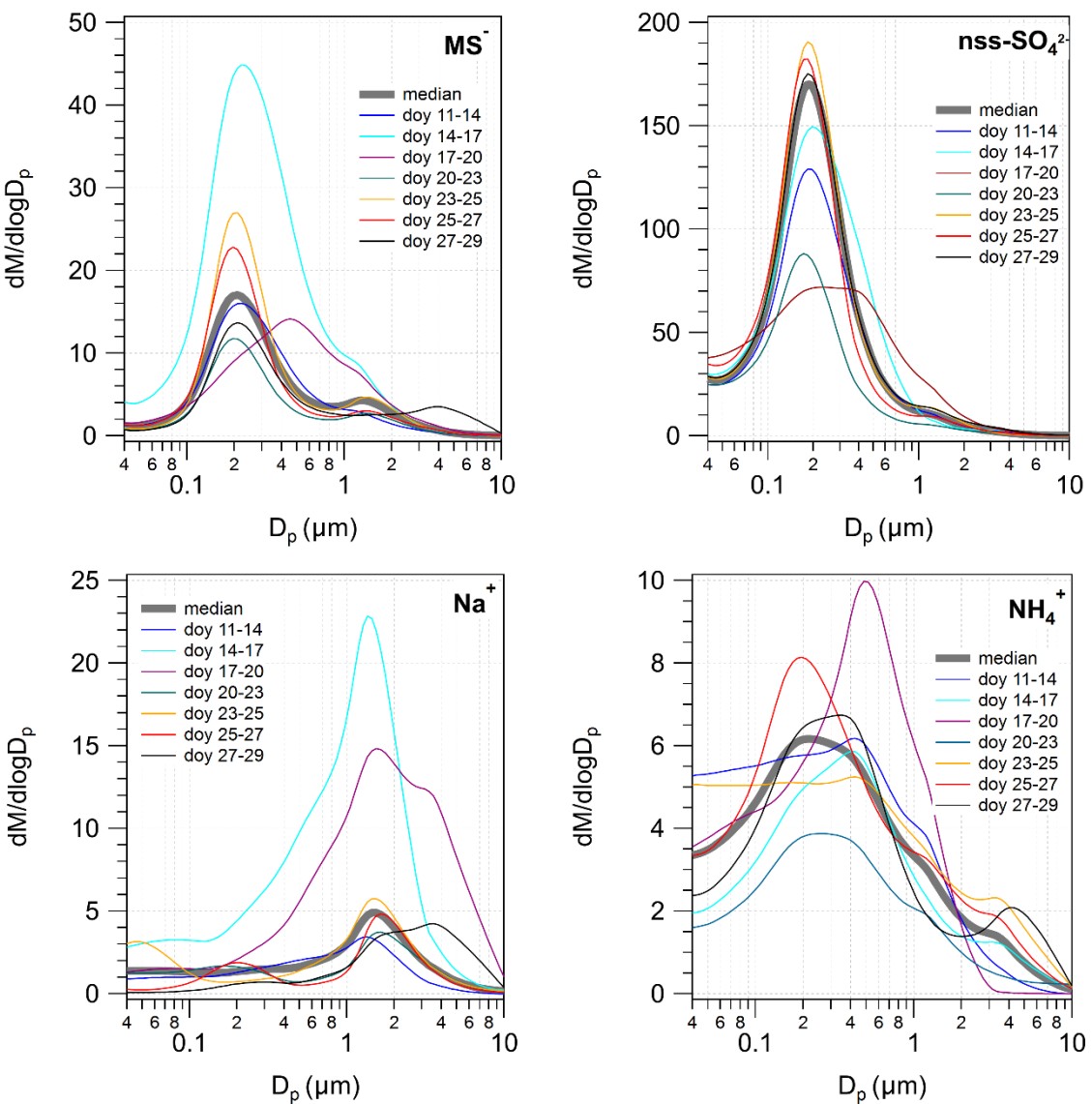

**Figure 12**. Results from size segregated (Berner impactor) sampling during the campaign 2016 with the following mass portion of the respective ions in the super-µm range (>1µm): MS⁻ 14% (a), nss-SO$_4^{2-}$ 3% (b), Na⁺ 46% (c), and NH$_4^+$ 15% (d). The bold gray line is the median mass size distribution of the corresponding ion.

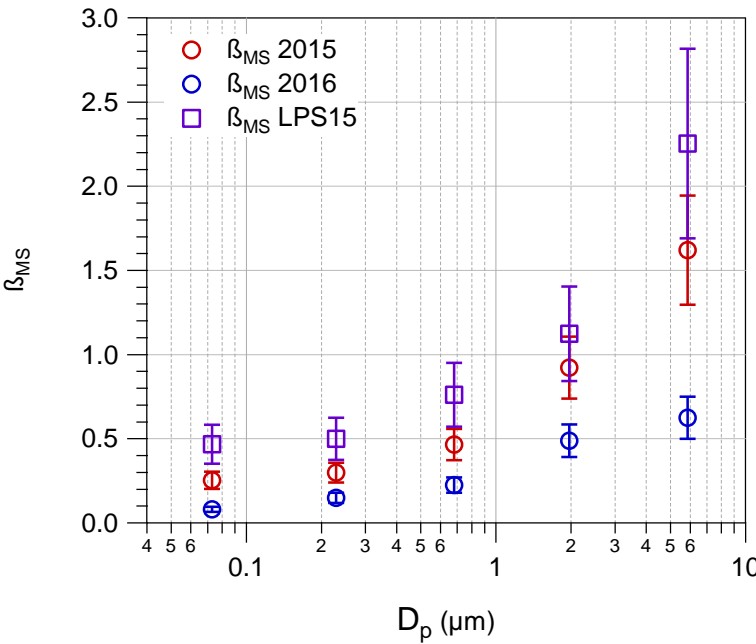

**Figure 13.** Mean size segregated results for the MS⁻/nss-SO₄²⁻ mass ratio $\text{ß}_{MS}$ determined for each of the 5 impactor stages for both seasons as well as for the event LPS15.

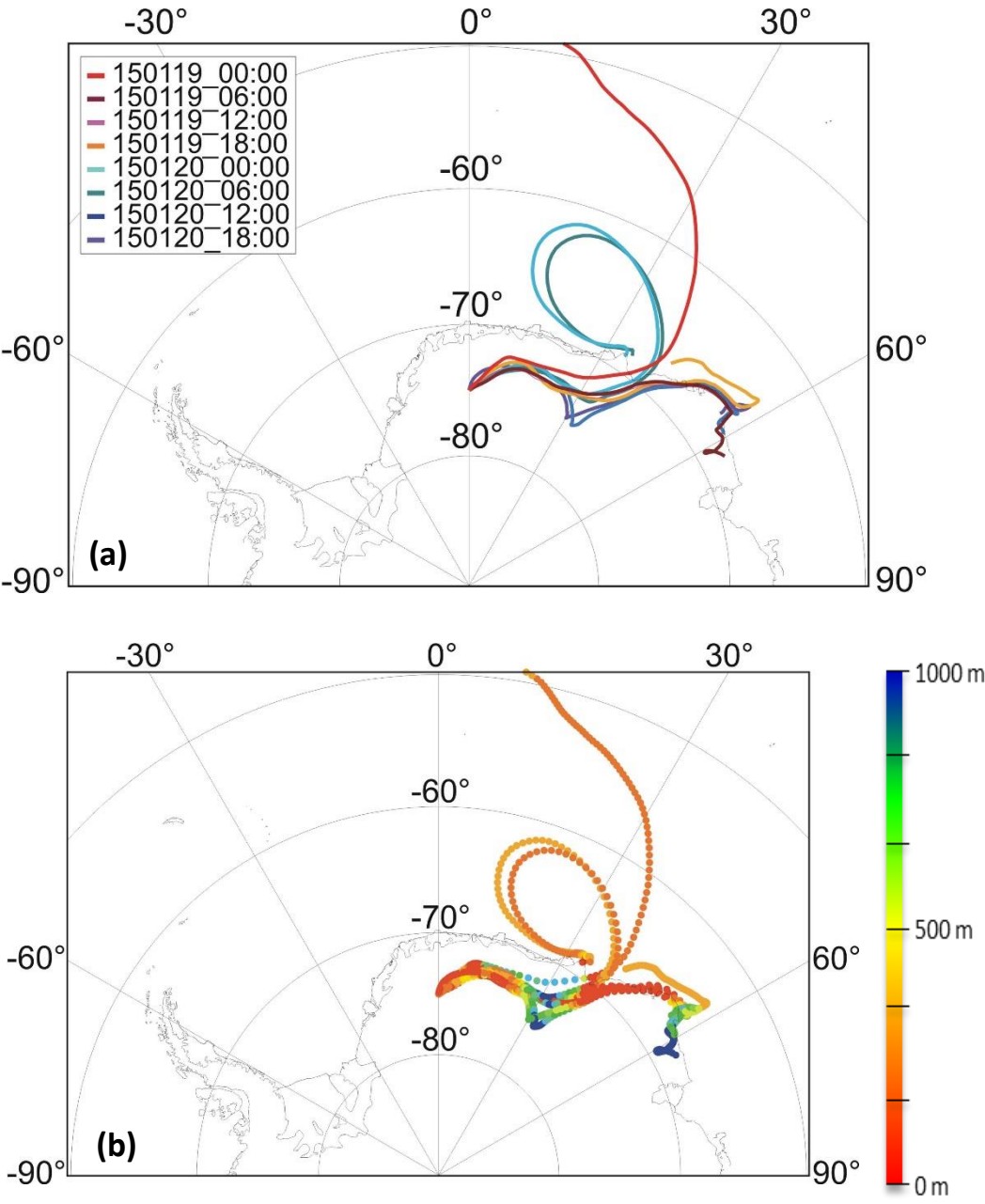

**Figure 14.** Five day backward trajectories during the NPF event, calculated with a trajectory starting height of 100 m above Kohnen at the points in time given in the legend (a). Below, the travel height above ground (local topography) is illustrated in a color coded scale (point interval 1 hour) (b).

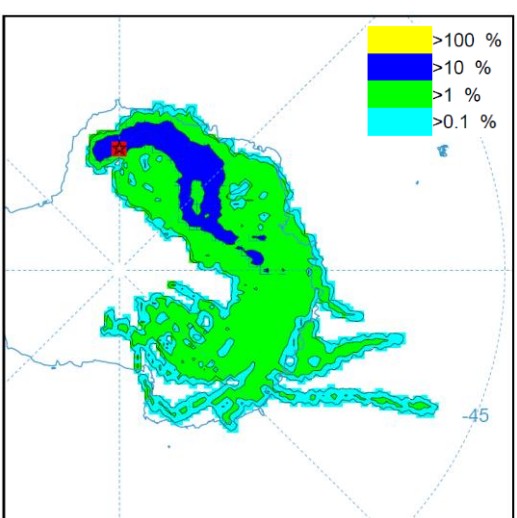

**Figure 15.** Daily 10-day backward trajectories (3D approach, starting height 100 m) during clear sky condition in 2016 (doy 12 to doy 31, N = 80). Shown is the relative (percentage) number of trajectory intersection on a given grid cell (resolution 1°×1°).

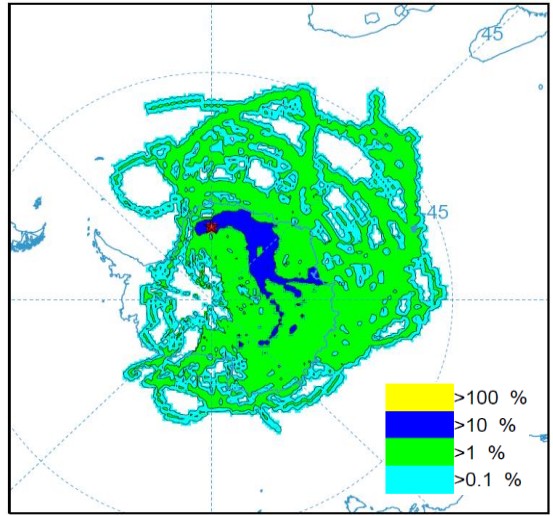

**Figure 16.** Daily 20-day backward trajectories (3D approach, starting height 100 m) during clear sky condition in 2016 (doy 12 to doy 31, N = 80). Shown is the relative (percentage) number of trajectory intersection on a given grid cell (resolution 1°×1°).