# Peer review of "Size distribution and size segregated ionic composition of the aerosol at the continental Antarctic site Kohnen (75°00'S, 00°04'E)"

_Atmospheric Chemistry and Physics, 2017_

## Referee Comment (RC1) · Anonymous Referee #2 · 20 Nov 2017

General comments:

The authors investigated the impact of different weather situations on the physical and chemical properties of the aerosols using measurements of aerosol number concentration, size distribution and size segregated aerosol sampling. They conclude that the marine aerosol is mostly transported and deposited through long-range transport during clear sky conditions. In addition, they report observation of formation and subsequent growth of aerosol particles traceable for 4-5 days and up to around 40-50 nm in size. This study is of particular interest for two reasons: i) papers comparing aerosols to larger scale meteorology in Antarctica are scarce and ii) this is the first

paper showing results of aerosol measurements from Kohnen. While this paper gives more insight into the aerosol processes in the Antarctica, at present it is lacking i) a better comparison to already existing papers, ii) more complete comparison to data from Neumayer and iii) throughout discussion and explanation of the NPF events observed during 17.-19.1.2015. These major issues need to be addressed before the publication in ACP. With these inclusions, this manuscript will be an important addition to the papers describing the aerosol chemical and physical properties and transport to Antarctica. Also, I am concerned about the coherency of the paper – binding the chapters (and maybe reconsidering their order and adding some new ones suggested in the specific comments) better together would make this manuscript more readable. Finally, and partly linked with the previous issue, the results, discussion and conclusions seem quite separate from each other. The authors need to find a way to build a story that is easier to follow and stress the importance of their findings.

Specific comments:

1. At page 3 in the last paragraph you refer to the summer concentration peak of biogenic sulphur aerosol but don't give an explanation. Please, add a short explanation for the clarity.

2. In chapter 2.2.3 you explain how you tried to validate the consistency of the HYSPLIT trajectories. Your second approach – varying the starting height between 10 and 100 m is not to my understanding very good approach due to the limited vertical representation of the model in such a small scale (at which height is Kohnen? And how well is the topography represented in HYSPLIT? The difference between 10 and 100 m might just not be visible at all in the model). Rather, use heights e.g. 100 m, 250 m, 500 m. However, HYSPLIT offers two easy, physically more justified ways to check the accuracy of the trajectories: ensemble and matrix trajectories. Due to the small number of days you could easily run those or one of the options to see whether there are discrepancies in the trajectory paths or not. Also, show somehow (give comparison values, example figure or so) how does the GDAS data differ from the isentropic

approximation?

3. Please include time labels (by 6 or 12 hours, or e.g. bigger labels by 12 hours and smaller by 6 hours) into your trajectories e.g. in Fig. 13A, S6A and S7A.

4. The authors are now mainly comparing their data with the meteorological conditions and to previous publications. It would be beneficial if you would include a small chapter of more throughout comparison of the aerosol chemical + physical composition between the different years (2015 and 2016). This is done already partly for the chemical composition but please, give some overall median / mean physical parameters for both years as well (e.g. particle number concentration, aerosol chemistry, . . .). You could make a comparison table including both the chemical and physical properties (e.g. for 2015 and 2016 overall and 2015 and 2016 LPS separately).

5. The second last paragraph of 4.1.1 is not easy to understand. Please, clarify.

6. The last paragraph in chapter 4.1.1 deals with comparison to Neumayer. This part is very important part, yet it is only 6 rows of text. You should do a more throughout comparison since the instrumentation in both Kohnen and Neumayer is very similar and one of the stations is much closer to the sea (N) while the other is further up on the continent (K). Please, extend your comparison as much as you can and also discuss on the meaning of the findings. How does the data from Neumayer support your conclusions?

7. In section 4.1.2, page 9, 1st paragraph, I think that the 20 hours needed for the particle growth is a slight overestimation or at least an upper limit. First of all, from the figure it seems that the NPF started at 10 nm or at lower sizes (if you look at the smallest observable size). Assuming that the particles would have reached e.g. 2 nm at the formation site (–> 10-2 = 8 nm of growth with advection), and assuming constant 8 m/s wind speed (from Figure 3), the time needed for the growth would be about 13 hours and the distance to the onset of formation less than 400 km. If the particle growth was even faster in the beginning (the authors note that organic vapors are needed to

explain the growth) and the particles would reach already 5 nm at the formation site, these values would reduce to 8.3 hours and 240 km. In either case, the formation must have taken place above the continent. Although it is important to compare the studies also to those made at the Antarctic plateau (the next paragraph), it would be needed to compare the results first to other Antarctic summer studies closer to the sea ice since the NPF and growth reported in this study compares better to those (see above and also those already included into the manuscript) and the airmasses according to the trajectories are not originated from the plateau.

8. The authors responded to the earlier comments that they included more references to the manuscript. I am however, concerned still about the discussion part of the manuscript and the references. They have referred very well to different studies of size segregated aerosol samplings around Antarctica but lack many major studies of aerosol physical properties close to Kohnen. Currently there are already quite a number of papers published from different stations around Dronning Maud Land (DML), close or further away from the coastline, under different meteorological conditions and in both marine or continental airmasses. All those papers draw their own part of the picture in the Antarctic aerosol. How do the authors of this manuscript relate their work and findings to the other papers? Below are listed few (not yet included) studies that this study could be compared with:

-Virkkula et al., JGR, 2006, doi:10.1029/2004JD004958 (Chemical aerosol NSD, Atlantic Ocean, Southern Ocean and Dronning Maud Land)

-Virkkula et al., Boreal Environ. Res., 2007, (PSD and NPF at Dronning Maud Land)

-Asmi et al., ACP, 2010, doi:10.5194/acp-10-4253-2010 (Aerosol hygroscopicity, chemical composition and NPF at Dronning Maud Land)

-Pant et al., JGR, 2010, http://dx.doi.org/10.1029/2009JD013481 (Aerosols during passage of cyclonic storms at Maitri)

-Yu and Luo, Atmosphere, 2010, doi:10.3390/atmos1010034 (Modelling study and comparisons to measurements, oceanic DMS and NPF around coastal Antarctica)

-Kyrö et al. ACP, 2013, doi:10.5194/acp-13-3527-2013 (NPF from continental biogenic precursors at Dronning Maud Land)

-Fiebig et al., ACP, 2014, doi:10.5194/acp-14-3083-2014 (Antarctic aerosol annual cycle at Dronning Maud Land)

-Chen et al., ACP, 2017, doi.org/10.5194/acp-17-13783-2017 (Air ion measurements at Dome C, including NPF event analysis, not from DML but since the authors compare their studies with an earlier paper from this station, this is included also)

9. Continuing the previous comment, Kyrö et al., 2013 presented very strong new particle formation events from Aboa, Dronning Maud Land (DML) and conclued that those events were associated with meltwater over the nunataks. Your trajectories during the NPF seem to come from the same direction as where they located the formation events to take place in NPF events that grew over several hours and it seems that some of the events they saw appeared during low pressure as well. Can you comment on this, how does your study compare with theirs? Also, the sulphuric acid concentration needed to explain the growth (1.4x10^7) compares surprisingly well to those concentrations reported in their study. I suggest that the authors read this study and comment on whether it is possible that the NPF observed at Kohnen is similar to what was reported by Kyrö et al.

10. From Figure 7 it looks like instead of one, you have observed 3 NPF events during consecutive days and every time there's first some local NPF during noon which disappears after couple of hours and then a new mode appears at around 10 nm at midnight. This is extremely interesting yet the authors don't seem to have fully noticed this or they are not discussing about this (the observation of two – not three – local NPF events is reported in chapter 3.2 but the discussion is completely missing). Please, explain this observation. And for the clarity it would be good if instead of Fig. 7 you have one figure

of particle NSD 17-21.1. so that the fine features of the SD evolution described above would be seen better, and one figure showing how the contamination at the station looks like (not needed). Also, your log-scale in the color code is not very clear. Instead of 0, 1, 2,... you could have 10ˆ0 or 1, 10ˆ1 or 10, 10ˆ2 or 100 and so on.

11. Furthermore, calculate GR, J3, J10, CS and CoagS for all of the 3 local (J3) and regional (J10) events.

12. How does the evolution of the aerosol size segregated chemistry during NPF event and subsequent growth support your conclusions?

13. It would be better if you combine figures 11 and S5 so that you have the medians for both 2015 and 2016 with very thick lines (e.g. 2015 with solid and 2016 with dashed lines) and then with thin solid and dashed lines with different colors the individual measurements. This way one would see the difference between the years easier; it is not very easy to compare the values or two figures when they are in different files. The days with particular interest (LPS) you could have with slightly thicker lines to stand out.

14. Please check the English in your manuscript – there are some inconsistencies or wrong words that change the meaning of the sentence completely (see detailed comments in "Technical corrections" –chapter) .

15. Finally, I think your manuscript is lacking a "storyline" – currently it is a set of chapters and paragraphs of results and discussion after each other but please, think how to glue and fit them together to make it an attractive story. Since your manuscript is rather short, you could consider combining the Results and Discussion sections together to make it more fluent to read. Please, stress the importance of your findings more. Right now the discussion is mostly concentrated on comparing to previous studies and it is not clearly stated what new your results bring to the understanding of the Antarctic atmosphere.

Technical corrections:

Page 1, Row 10: "an outstanding low pressure system" – as per Oxford English dictionary, oustanding means either "exceptionally good" or "not yet resolved". Neither of these meanings fit very well to a low pressure system. How about changing it to remarkable / striking / notable / exceptional?

Page 1, Row 21: "aerosol characterized by an usually mono-modal..." Do you mean "by unusually mono-modal..." or "by usually mono-modal..."

Page 2, Row 7: "under predict" to "underpredict"

Page 2, Row 13: Your reference to Raes et al., 2000 is already rather old. Consider adding also more references.

Page 4, Row 23: "all up" do you mean "in total"?

Page 4, Row 27: "we have to", is in present tense while your previous sentence is in imperfect. Change to "we had to"

Page 6, Row 17: "provoked by local combustion were" –> "provoked by local combustion are"

Page 7, Row 9: "outstanding Na+", consider changing to "exceptionally high Na+". Also, again in Page 7, Row 17, some other word would fit better than "outstanding".

Page 7, Row 33: Difficult sentence. Change to e.g. "In Dronning Maud Land, cyclone driven marine air intrusions are infrequent, sporadic events that are often associated with high-precipitation rates (refs)".

Page 8, Row 22: as well as –> and; short stormy period –> a short stormy period

Page 8, Row 30: Verisimilar –> Very similar

Page 8, Row 38: "weather situation had influenced transport of marine aerosol..." –> "weather situation transported marine aerosol..."

Page 9, Row 10: "magnitude to high" to "magnitude too high"

Page 11, Row 5: Condition written two times; "clear sky condition conditions" –> "clear sky condition"

Page 11, Row 13: "understand the role biogenic aerosol. . ." –> "understand the role of biogenic aerosol. . ."
* * *

---

## Referee Comment (RC2) · Anonymous Referee #1 · 21 Nov 2017

General comments

The paper presents the result on the complex link between size distribution, ionic composition and new particle formation episodes at Kohnen station (Antarctica). Backward trajectories analysis was accomplished in order to understand air mass source areas and traveling routes. The paper is very important in understanding the present day transport and deposition processes and factors influencing new particle formation. Besides, such studies are fundamental to interpret past climatic and environmental change from ice core stratigraphies especially in this site were one of the two deep ice cores within the EPICA project in Dronning Maud Land is obtained. The paper is in my

opinion well organized, the results on the considered events are well constrained, and it deserve the publication on ACP. Only some technical English corrections are needed.

---

## Author Comment (AC1) · 21 Dec 2017

**Response to Reviewer 2:**

First of all we would like to thank reviewer#2 for his effort in evaluating our manuscript (ms)! Her/his thorough evaluation admittedly revealed several shortcomings in our manuscript. According to the comments, we reconsidered and rectified our ms. Below, we give a point to point reply. For convenience and to avoid an unnecessary inflation of this response letter, we marked the corresponding changes in our ms in yellow and refrain from listing all revised fragments. Revised fragments presented here are straight below our response *("in quotation marks and in italics")*.

*General comments: The authors investigated the impact of different weather situations on the physical and chemical properties of the aerosols using measurements of aerosol number concentration, size distribution and size segregated aerosol sampling. They conclude that the marine aerosol is mostly transported and deposited through long-range transport during clear sky conditions. In addition, they report observation of formation and subsequent growth of aerosol particles traceable for 4-5 days and up to around 40-50 nm in size. This study is of particular interest for two reasons: i) papers comparing aerosols to larger scale meteorology in Antarctica are scarce and ii) this is the first paper showing results of aerosol measurements from Kohnen. While this paper gives more insight into the aerosol processes in the Antarctica, at present it is lacking i) a better comparison to already existing papers, ii) more complete comparison to data from Neumayer and iii) throughout discussion and explanation of the NPF events observed during 17.-19.1.2015. These major issues need to be addressed before the publication in ACP. With these inclusions, this manuscript will be an important addition to the papers describing the aerosol chemical and physical properties and transport to Antarctica. Also, I am concerned about the coherency of the paper – binding the chapters (and maybe reconsidering their order and adding some new ones suggested in the specific comments) better together would make this manuscript more readable. Finally, and partly linked with the previous issue, the results, discussion and conclusions seem quite separate from each other. The authors need to find a way to build a story that is easier to follow and stress the importance of their findings.*

*Specific comments:*

*1. At page 3 in the last paragraph you refer to the summer concentration peak of biogenic sulphur aerosol but don't give an explanation. Please, add a short explanation for the clarity.*

We now mention that the distinct summer maximum of biogenic sulfur aerosol is provoked by the seasonality of the marine biogenic activity of the surrounding ocean.

*"This extended synoptic documentation of physics and chemistry of Antarctic aerosol primarily concentrates here on biogenic sulfur aerosol, due to its distinct seasonal summer concentration peak caused by the seasonality of marine biogenic activity of the surrounding Southern Ocean (Weller and Wagenbach, 2007)."*

*2. In chapter 2.2.3 you explain how you tried to validate the consistency of the HYSPLIT trajectories. Your second approach – varying the starting height between 10 and 100 m is not to my understanding very good approach due to the limited vertical representation of the model in such a small scale (at which height is Kohnen? And how well is the topography represented in HYSPLIT? The difference between 10 and 100 m might just not be visible at all in the model). Rather, use heights e.g. 100 m, 250 m, 500 m. However, HYSPLIT offers two easy, physically more justified ways to check the accuracy of the trajectories: ensemble and matrix trajectories. Due to the small number of days you could easily run those or one of the options to see whether there are discrepancies in the trajectory paths or not.*

*Also, show somehow (give comparison values, example figure or so) how does the GDAS data differ from the isentropic approximation?*

*3. Please include time labels (by 6 or 12 hours, or e.g. bigger labels by 12 hours and smaller by 6 hours) into your trajectories e.g. in Fig. 13A, S6A and S7A.*

First of all we modified our trajectory figures to facilitate the readability. We now calculated five day backward trajectories ensembles during the NPF event (Supplementary Material Fig. S.9; starting points were varied by ±1° longitude and ±1° latitude each, while for the height above ground 0 m, 250 m, and 500 m were chosen, respectively). In addition, we present the results of five day backward trajectories based on isentropic instead of 3D approach for the NPF event (Supplementary Material Fig. S.10) and provided weather charts for LPS15 and LPS16 (Supplementary Material Figs. S.2 and S.3). We usually varied the starting height level between 10 m and 100 m because this is typically the width of the stable boundary layer. Note that trajectories started well above the stable boundary layer would most probably not be very representative for ground level measurements. The altitude of Kohnen was already given in Fig. 1 but now repeated in the main text (Chapter 2.1).

*"(iii) trajectories ensembles whose starting points were varied by ±1° longitude and ±1° latitude each, while for the height above ground 0 m, 250 m, and 500 m were chosen, respectively. The impact of these different initial conditions on our conclusions will be appraised in the Discussion section. Finally weather charts generated by the Antarctic Mesoscale Prediction System (AMPS) were used to assess the general weather situation (Powers et al., 2003; http://www2.mmm.ucar.edu/rt/amps/information/amps_esg_data_info.html, last access: 27 October 2017)."*

*4. The authors are now mainly comparing their data with the meteorological conditions and to previous publications. It would be beneficial if you would include a small chapter of more throughout comparison of the aerosol chemical + physical composition between the different years (2015 and 2016). This is done already partly for the chemical composition but please, give some overall median / mean physical parameters for both years as well (e.g. particle number concentration, aerosol chemistry, . . .). You could make a comparison table including both the chemical and physical properties (e.g. for 2015 and 2016 overall and 2015 and 2016 LPS separately).*

We added a corresponding additional table (Tab. 2) and refer to the results in Chapter 3.2 and 4.1.1.

*5. The second last paragraph of 4.1.1 is not easy to understand. Please, clarify.*

We agree, amongst others there is a bad example for a nested sentence! We changed the wording in the second last paragraph of 4.1.1.

*"We may speculate that now intrusions of marine boundary layer into the so-called buffer layer were responsible for efficient advection of gaseous DMS photo oxidation products like $SO_2$ and DMSO (Davis et al., 1998; Russel et al., 1998). Russel et al. (1998) assumed that the buffer layer typically extends from the turbulent marine boundary layer (400 m to 700 m) up to a capping inversion (1400 m to 1900 m). While transported to continental Antarctica, gas phase photo oxidation processes should have dominated, leading finally to a preferred formation of $H_2SO_4$ at the expense of MSA (Preunkert et al., 2008). Anyway, this plausible but subtle transport route could not be unequivocally deduced from respecting backward trajectory analyses, because in the case at hand, the presence and the extent of a buffer layer could not be ascertained from available meteorological data."*

*6. The last paragraph in chapter 4.1.1 deals with comparison to Neumayer. This part is very important part, yet it is only 6 rows of text. You should do a more throughout comparison since the instrumentation in both Kohnen and Neumayer is very similar and one of the stations is much closer to*

*the sea (N) while the other is further up on the continent (K). Please, extend your comparison as much as you can and also discuss on the meaning of the findings. How does the data from Neumayer support your conclusions?*

The reviewer addresses here a somewhat delicate issue: One of the ms authors (RW) actually evaluated the impact of cyclones on aerosol transport to Neumayer based on more than 20 years continuous CP and ionic composition data from this site (not pubished). Unfortunately, the results emerged not as clear-cut compared to similar pubished studies (which are based on much shorter time series) from other sites might suggest (e.g. Pant et al. 2010): At Neumayer, a significant impact of "typical" cyclone passages on aerosol load is very often virtually absent. In the remaining cases, particle number concentrations and ionic concentration changes reached their (rarely coinciding) maxima alternately just before, during, and in the aftermath of a given cyclone passage. Surely, these findings would deserve a thorough discussion of its own, which is clearly beyond the scope of this ms! On these grounds we restrict the comparison with Neumayer data for the period of the concurrent Kohnen campaign. Note, that during this period, SMPS and impactor measurements were not available for Neumayer (actually we temporarily removed these instruments for deployment at Kohnen!). Nevertheless, we somewhat extended the discussion of the Neumayer data (Chapter 4.1.1).

*"Consequently we can assume that this characteristic weather situation transported marine aerosol throughout DML. But in contrast to Kohnen the impact of LPS15 on the delayed particle concentration and ionic composition maxima was less pronounced. Pant et al. (2010) concluded from their particle concentration and size distribution data measured at coastal Maitri that during the impact of cyclones, coarse mode sea salt aerosol increased by an order of magnitude compared to calm weather conditions, similar to our results during LPS15 (Table 2)."*

*7. In section 4.1.2, page 9, 1st paragraph, I think that the 20 hours needed for the particle growth is a slight overestimation or at least an upper limit. First of all, from the figure it seems that the NPF started at 10 nm or at lower sizes (if you look at the smallest observable size). Assuming that the particles would have reached e.g. 2 nm at the formation site (–> 10-2 = 8 nm of growth with advection), and assuming constant 8 m/s wind speed (from Figure 3), the time needed for the growth would be about 13 hours and the distance to the onset of formation less than 400 km. If the particle growth was even faster in the beginning (the authors note that organic vapors are needed to explain the growth) and the particles would reach already 5 nm at the formation site, these values would reduce to 8.3 hours and 240 km. In either case, the formation must have taken place above the continent. Although it is important to compare the studies also to those made at the Antarctic plateau (the next paragraph), it would be needed to compare the results first to other Antarctic summer studies closer to the sea ice since the NPF and growth reported in this study compares better to those (see above and also those already included into the manuscript) and the airmasses according to the trajectories are not originated from the plateau.*

We think that our declaredly crude estimate is based on realistic assumptions. We agree with the reviewer that our estimate is most probably an upper limit but we relied on back trajectories instead of simply assuming a constant wind velocity to assess the distance to the onset of the particle formation. Indeed the GR determined in this studies seem to compare better to values from coastal site (which is now mentioned in the revised ms), but please note that particle growth rates determined for Antarctic NPF are within a (very) wide range, regardless whether they were measured on the plateau or close to the coast! In the revised ms we appreciably extended the discussion on NPF and GR including relevant studies from other sites (Chapter 4.1.2).

*"Interestingly, Kyrö et al. (2013) identified biogenic emissions by nearby melting ponds as a potential source for condensable vapour. The surroundings of Kohnen, however, are completely ice covered throughout as typical for the Antarctic Plateau region. The nearest rocky outcrops are more than 200 km away. [...] Furthermore, GR*

*determined so far at coastal sites (Virkkula et al., 2007; Asmi et al., 2010; Kyrö et al., 2013; Weller et al., 2015) appeared comparable to GR reported from continental Antarctica (Järvinen et al., 2013; Chen et al., 2017) and were within a similar broad range between some 0.2 nm h⁻¹ up to 8.8 nm h⁻¹.*

*The impact of passing cyclones on particle size distributions was up to now just studied by Pant et al. (2010) at coastal Maitri, These authors observed bimodal PNSD with a coarse mode maxima around 2 µm and a broad Aitken mode between 0.04 µm and 0.1 µm when a storm approached the site. Occasionally NPF occurred just after the passage of a cyclone associated with particle growth rates between 0.2 and 0.6 nm h⁻¹. From meteorological data these authors conclude that the observed NPF events were linked with mixing of marine and continental during subsidence of free tropospheric air after the storm (Pant et al., 2010).*

*8. The authors responded to the earlier comments that they included more references to the manuscript. I am however, concerned still about the discussion part of the manuscript and the references. They have referred very well to different studies of size segregated aerosol samplings around Antarctica but lack many major studies of aerosol physical properties close to Kohnen. Currently there are already quite a number of papers published from different stations around Dronning Maud Land (DML), close or further away from the coastline, under different meteorological conditions and in both marine or continental airmasses. All those papers draw their own part of the picture in the Antarctic aerosol. How do the authors of this manuscript relate their work and findings to the other papers? Below are listed few (not yet included) studies that this study could be compared with:*

*-Virkkula et al., JGR, 2006, doi:10.1029/2004JD004958 (Chemical aerosol NSD, Atlantic Ocean, Southern Ocean and Dronning Maud Land)*
*-Virkkula et al., Boreal Environ. Res., 2007, (PSD and NPF at Dronning Maud Land) -Asmi et al., ACP, 2010, doi:10.5194/acp-10-4253-2010 (Aerosol hygroscopicity, chemical composition and NPF at Dronning Maud Land)*
*-Pant et al., JGR, 2010, http://dx.doi.org/10.1029/2009JD013481 (Aerosols during passage of cyclonic storms at Maitri)*
*-Yu and Luo, Atmosphere, 2010, doi:10.3390/atmos1010034 (Modelling study and comparisons to measurements, oceanic DMS and NPF around coastal Antarctica)*
*-Kyrö et al. ACP, 2013, doi:10.5194/acp-13-3527-2013 (NPF from continental biogenic precursors at Dronning Maud Land) -Fiebig et al., ACP, 2014, doi:10.5194/acp-14-3083-2014 (Antarctic aerosol annual cycle at Dronning Maud Land)*
*-Chen et al., ACP, 2017, doi.org/10.5194/acp-17-13783-2017 (Air ion measurements at Dome C, including NPF event analysis, not from DML but since the authors compare their studies with an earlier paper from this station, this is included also)*

Thanks a lot for calling our attention to the paper by Pant et al. (2010)! Actually, we were not aware of this interesting and elaborate study, which seems obviously rarely considered in concerning publications! We refer now to this paper in the Introduction and Discussion sections (especially Chapter 4.1.2) of our revised ms. Apart from that, we are well aware of the remaining publications listed above. We think that in the revised ms all relevant publications are now adequately included in our discussion.

*9. Continuing the previous comment, Kyrö et al., 2013 presented very strong new particle formation events from Aboa, Dronning Maud Land (DML) and conclued that those events were associated with meltwater over the nunataks. Your trajectories during the NPF seem to come from the same direction as where they located the formation events to take place in NPF events that grew over several hours and it seems that some of the events they saw appeared during low pressure as well. Can you comment on this, how does your study compare with theirs? Also, the sulphuric acid concentration needed to explain the growth (1.4x10ˆ7) compares surprisingly well to those concentrations reported in their study. I suggest that the authors read this study and comment on whether it is possible that the NPF observed at Kohnen is similar to what was reported by Kyrö et al.*

Again, we were well aware of the mentioned study and we considered it now in the Discussion (Chapter 4.1.2). However, though this publication presents very interesting and challenging results (for the first time a NPF source located in continental Antarctica could be substantiated), we definitely exclude the impact of melting pots on our NPF events. Such areas are several hundred of km away from Kohnen and their potential effect has to be set in relation with the overwhelming marine biogenic source which is also just some hundred km away from Kohnen.

*10. From Figure 7 it looks like instead of one, you have observed 3 NPF events during consecutive days and every time there's first some local NPF during noon which disappears after couple of hours and then a new mode appears at around 10 nm at midnight. This is extremely interesting yet the authors don't seem to have fully noticed this or they are not discussing about this (the observation of two – not three – local NPF events is reported in chapter 3.2 but the discussion is completely missing). Please, explain this observation. And for the clarity it would be good if instead of Fig. 7 you have one figure of particle NSD 17-21.1. so that the fine features of the SD evolution described above would be seen better, and one figure showing how the contamination at the station looks like (not needed). Also, your log-scale in the color code is not very clear. Instead of 0, 1, 2,. . . you could have 10ˆ0 or 1, 10ˆ1 or 10, 10ˆ2 or 100 and so on.*

We agree with the reviewer in this respect. In fact, we described the observed NPF and particularly the distinct features before this event in a rather superficial manner. According to the reviewers comment, we revised Chapter 3.2, characterized all observed features (for this we added a new Table 3), and presented a modified contour plot of the NPF (Fig. 7; also log-scale is now changed).

*"Table 2 summarizes particle concentrations as well as ionic composition of the aerosol during the impact of LPS15, LPS16, and clear sky conditions. While during 2015 CPC and UCP concentrations were by nearly an order of magnitude higher compared to clear sky conditions, the effect of LPS16 was not obvious. [...] From their temporal evolution we derived a continuous growth rate of $0.6\pm0.08$ nm $h^{-1}$ for the whole event between doy 19 02:50 and doy 20 24:00. In addition, a separate calculation for the first and second part of the event (boundary: noon of doy 19) resulted in virtually idendical growth rates of $0.6\pm0.1$ nm $h^{-1}$ for both sections. [...] In addition, we observed enhanced UCP concentrations between 10 nm and 25 nm during doy 18 in the morning hours, just before the actual NPF, and in the evening of doy 19 (numbered 2, 4, and 5 in Fig. 7). Apart from that, discernible natural nucleation bursts occurred around noon of doy 17 and 18 (numbered 1 and 3 in Fig. 7). All these transient UCP maxima did not show any detectable particle growth. The nucleation bursts were characterized by increasing particle concentrations from slightly above 5 nm downward towards the lower instrumental size limit ("open" distribution), indicating local nucleation. Table 3 provides a summary of the respecting particle formation rates $J_{3-25}$ and the range of the observed particle diameter $D_P$ for these events. In contrast, local contamination provoked strong particle bursts which typically showed spiky and strongly enhanced particle concentrations ($UCP_{3-25}$ concentrations $>2500$ $cm^{-3}$, $J_{3-25}$ typically $>10$ $s^{-1}$) within a wide particle size range as marked with white frames in Supplementary Material Fig. S4."*

*11. Furthermore, calculate GR, J3, J10, CS and CoagS for all of the 3 local (J3) and regional (J10) events.*

Where possible, we provided now additional information for the mentioned particle bursts (Table 3). As already noted in the original ms, GR could only be determined for the main event.

*12. How does the evolution of the aerosol size segregated chemistry during NPF event and subsequent growth support your conclusions?*

It is obvious that the mentioned NPF was ultimately provoked by marine biogenic emissions, which is clearly stated the Results and Discussion chapters. Aerosol bulk and size segregated ionic composition in terms of NPF are discussed now (amongst others) in the revised Chapter 4.1.1. As already stated,

peak biogenic sulfur **mass concentrations** occurred in final stage of LPS15 and the NPF event. The contribution of the nucleated particles (high particle **number concentrations** within the nucleation and Aitken mode) to the total biogenic sulfur (mass) concentrations is expectedly low (see also Weller et al. 2015, Fig. 1e therein), due to the $(D_p)^3$ dependence of volume or mass. To be specific: Similar to Weller et al. (2015) we calculated from the measured nano-DMA size distribution spectra the total aerosol mass concentration between 3 nm and 64 nm ($M_{DMA3-64}$), assuming a particle density of 1.8 g cm$^{-3}$ und compared the results with biogenic sulfur concentrations derived from respecting TNy samples ($M_{bioS}$). During NPF (doy 17 through doy 20) $M_{DMA3-64}$ corresponded to only 26 ng m$^{-3}$ compared $M_{bioS}$ = 208 ng m$^{-3}$. In a nutshell: The chemical composition of the nucleated particles cannot be assessed by the employed low volume or impactor sampling methods (note the lower cut–off of around 42 nm and the very poor temporal resolution of our impactor sampling). For this purpose sophisticated aerosol mass spectrometer measurements would be necessary as first described by Giordano et al. (2017) for an Antarctic site (doi: 10.5194/acp-17-1-2017).

*13. It would be better if you combine figures 11 and S5 so that you have the medians for both 2015 and 2016 with very thick lines (e.g. 2015 with solid and 2016 with dashed lines) and then with thin solid and dashed lines with different colors the individual measurements. This way one would see the difference between the years easier; it is not very easy to compare the values or two figures when they are in different files. The days with particular interest (LPS) you could have with slightly thicker lines to stand out.*

We changed the respecting figures according to the reviewers suggestions and moved Fig. S.5 from the Supplemenatry Material to the main ms (now Fig. 12). Note that in case of 2016 we could not assign particular impactor results to LPS16 due to the short duration of this event compared to the sampling period.

*14. Please check the English in your manuscript – there are some inconsistencies or wrong words that change the meaning of the sentence completely (see detailed comments in "Technical corrections" – chapter) .*

See below.

*15. Finally, I think your manuscript is lacking a "storyline" – currently it is a set of chapters and paragraphs of results and discussion after each other but please, think how to glue and fit them together to make it an attractive story. Since your manuscript is rather short, you could consider combining the Results and Discussion sections together to make it more fluent to read. Please, stress the importance of your findings more. Right now the discussion is mostly concentrated on comparing to previous studies and it is not clearly stated what new your results bring to the understanding of the Antarctic atmosphere.*

We rearranged and considerably revised the whole Discussion section. We hope that these substantial revisons conform to the above mentioned concern of the reviewer. Notwithstanding, we refrain from merging Results and Discussion into one chapter. A separation still seems to us more suitable

*Technical corrections:*

*Page 1, Row 10: "an outstanding low pressure system" – as per Oxford English dictionary, oustanding means either "exceptionally good" or "not yet resolved". Neither of these meanings fit very well to a low pressure system. How about changing it to remarkable / striking / notable / exceptional?*

We changed the expression "outstanding" according to the reviewers recommendation.

*Page 1, Row 21: "aerosol characterized by an usually mono-modal. . ." Do you mean "by unusually mono-modal. . ." or "by usually mono-modal. . ."*

*The latter is correct; we deleted "an".*

*Page 2, Row 7: "under predict" to "underpredict"*

*Corrected.*

*Page 2, Row 13: Your reference to Raes et al., 2000 is already rather old. Consider adding also more references.*

*We added a more recent publication*

*Page 4, Row 23: "all up" do you mean "in total"?*

*Corrected.*

*Page 4, Row 27: "we have to", is in present tense while your previous sentence is in imperfect. Change to "we had to"*

*Corrected.*

*Page 6, Row 17: "provoked by local combustion were" –> "provoked by local combustion are"*

*Corrected.*

*Page 7, Row 9: "outstanding Na+", consider changing to "exceptionally high Na+". Also, again in Page 7, Row 17, some other word would fit better than "outstanding".*

*Corrected.*

*Page 7, Row 33: Difficult sentence. Change to e.g. "In Dronning Maud Land, cyclone driven marine air intrusions are infrequent, sporadic events that are often associated with high-precipitation rates (refs)".*

*Corrected.*

*Page 8, Row 22: as well as –> and; short stormy period –> a short stormy period*

*Corrected.*

*Page 8, Row 30: Verisimilar –> Very similar*

*We meant "plausible" (now applied) which is, according to our dictionary, synonym to verisimilar.*

*Page 8, Row 38: "weather situation had influenced transport of marine aerosol. . ." –> "weather situation transported marine aerosol. . ."*

*Corrected.*

*Page 9, Row 10: "magnitude to high" to "magnitude too high"*

*Corrected.*

*Page 11, Row 5: Condition written two times; "clear sky condition conditions" –> "clear sky condition"*

*Corrected.*

*Page 11, Row 13: "understand the role biogenic aerosol. . ." –> "understand the role of biogenic aerosol.*

*Corrected.*